# Beyond Gemini-3-Pro: Revisiting LLM Routing and Aggregation at Scale

**Shengji Tang** [* 1 2]  **Weihao Lin** [* 1 3]  **Peng Ye**[✉ 1 2]  **Jingqi Ye** [4]  **Hao Li** [1 5]  **Yiqun Zhang** [1 6]  **Xiaosong Wang** [1]
**Bo Zhang** [1]  **Shuyue Hu** [1]  **Tao Chen** [3]  **Lei Bai** [1]  **Wanli Ouyang** [1 2]

## Abstract

Large Language Models (LLMs) have rapidly advanced, with Gemini-3-Pro setting a new performance milestone. In this work, we explore collective intelligence as an alternative to monolithic scaling, and demonstrate that open-source LLMs' collaboration can surpass Gemini-3-Pro. We first revisit LLM routing and aggregation at scale and identify three key bottlenecks: (1) current train-free routers are limited by a query-based paradigm focusing solely on textual similarity; (2) recent aggregation methods remain largely static, failing to select appropriate aggregators for different tasks; (3) the complementarity of routing and aggregation remains underutilized. To address these problems, we introduce JiSi, a novel framework designed to release the full potential of LLMs' collaboration through three innovations: (1) Query-Response Mixed Routing capturing both semantic information and problem difficulty; (2) Support-Set-based Aggregator Selection jointly evaluating the comprehensive and domain capacity of aggregators; (3) Adaptive Routing-Aggregation Switch dynamically leveraging the advantages of routing and aggregation. Comprehensive experiments on nine benchmarks demonstrate that JiSi can surpass Gemini-3-Pro with only 47% costs by orchestrating ten open-source LLMs, while outperforming mainstream baselines. It suggests that collective intelligence represents a novel path towards Artificial General Intelligence (AGI).

---

[*]Equal contribution  [1]Shanghai AI Lab [2]The Chinese University of Hong Kong [3]Fudan University [4]University of Science and Technology of China [5]Northwestern Polytechnical University [6]Northeastern University. Correspondence to: Peng Ye <yepeng@pjlab.org.cn>.

*Proceedings of the 43ʳᵈ International Conference on Machine Learning*, Seoul, South Korea. PMLR 306, 2026. Copyright 2026 by the author(s).

## 1. Introduction

The landscape of Artificial Intelligence (AI) has been fundamentally reshaped by Transformer-based Large Language Models (LLMs) (Vaswani et al., 2017; Brown et al., 2020). By continuously scaling up model parameters and training data, LLMs have achieved unprecedented capabilities (Kaplan et al., 2020; Comanici et al., 2025; Anthropic, 2025a; OpenAI, 2025a;b; Google, 2025). Most notably, **Gemini-3-Pro** (Google, 2025) has established a significant performance milestone across different fields, including knowledge and reasoning, coding and engineering, general chat and factuality. The overwhelming success of Gemini-3-Pro further reinforces the prevailing paradigm of training increasingly powerful monolithic models. However, this naturally raises a fundamental question: is the indefinite scaling of a monolithic "super model" the only path towards Artificial General Intelligence (AGI)?

In this work, we instead explore collective intelligence as an alternative path, demonstrating that the collaboration of individually weaker open-source LLMs can surpass leading closed-source models, including Gemini-3-Pro. To achieve this goal, we first revisit routing and aggregation at scale, including giant LLMs and challenging benchmarks. We find that: (1) Current routing methods (Shnitzer et al., 2023; Chen et al., 2024b; Zhang et al., 2025a) typically project queries into a latent space via embedding models for clustering or matching-based model selection. However, pre-trained embedding models only capture text similarity and fail to perceive deep semantics or task difficulty, leading to suboptimal or mismatched selection. (2) Recent aggregation techniques (Wang et al., 2024a; Li et al., 2025b;a) mainly exploit complementary information but are largely static, making it difficult to adaptively select aggregators with strong aggregation and domain-specific capabilities, which inevitably suffer from bottlenecks in critical response generation. (3) The complementarity between routing and aggregation is overlooked: routing offers stability but is constrained by a single model, aggregation can exceed individual model limits yet is vulnerable to low-quality outputs and noise (Li et al., 2025a; Wolf et al., 2025).

To break through the above bottlenecks, we propose a novel

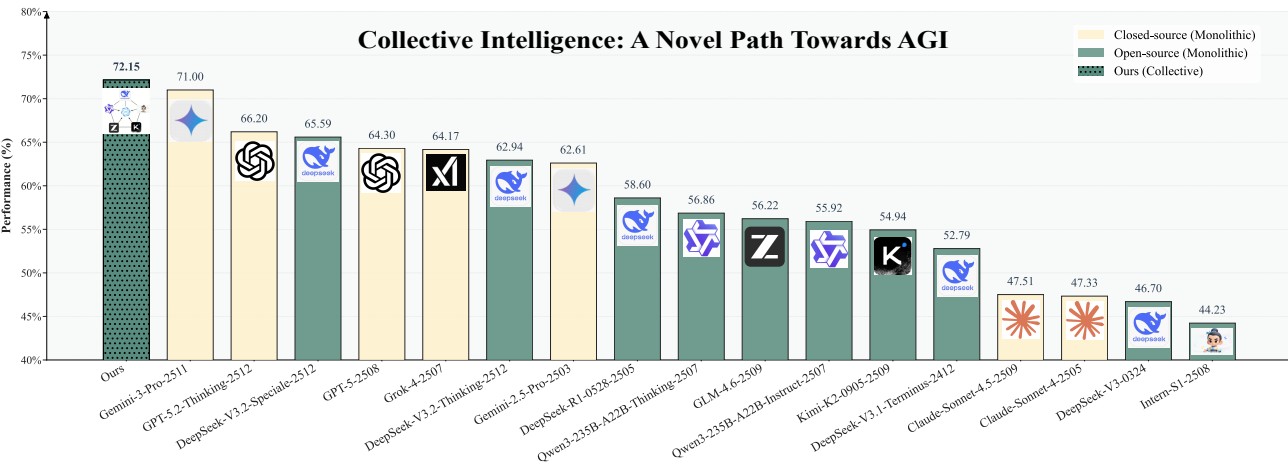

*Figure 1.* By orchestrating open-source LLMs collectively, JiSi surpasses all monolithic LLMs, including leading closed-source LLMs such as Gemini-3-Pro, suggesting that collective intelligence represents a novel path towards AGI.

framework entitled **JiSi**[1], featuring a minimalist but surprisingly effective design. Specifically, based on a pre-built embedding bank, it contains three core techniques: 1) **Query-Response Mixed Routing**: To capture deep semantics and task difficulty beyond shallow text embeddings, we utilize LLM-generated response embeddings to refine the routing process, causing a more precise match between input queries and LLMs. 2) **Support-Set-based Aggregator Selection**: To strike a balance between domain-specific and general aggregation capabilities, we leverage prior scores derived from a larger-scale embedding support set to select suitable aggregators dynamically. 3) **Adaptive Routing-Aggregation Switch**: We introduce an adaptive switch mechanism that selectively aggregates excellent expert responses or switches to a pure routing strategy according to their refined prior scores. This strategy suppresses noise and prevents the propagation of inferior information to the aggregator.

We conduct comprehensive experimental comparisons and analyses to assess the effectiveness of the proposed method JiSi. Specifically, we select ten open-source giant LLMs as candidate models and evaluate JiSi across nine challenging benchmarks. As shown in Fig. 1 and Table 2, the proposed JiSi surpasses all current state-of-the-art closed-source proprietary models. Compared to Gemini-3-Pro, it achieves an average performance gain of **+1.15** and saves **53.23%** cost. As shown in Fig. 2, JiSi can take specific advantages of candidate LLMs and break through their upper bound. In addition, JiSi remarkably outperforms all mainstream baselines. Furthermore, we verify that JiSi, as a concise training-free framework, exhibits consistent performance improvement with the introduction of new LLMs and tasks,

which highlights its strong scalability. It suggests that collective intelligence is a potential new path towards AGI.

## 2. Related works

This paper investigates collective intelligence in multi-LLM collaboration, which can be broadly divided into router-based and aggregation-based methods.

**Router-based Methods**. These approaches exploit the domain-specific strengths of heterogeneous models by routing queries based on task requirements. RouterDC (Chen et al., 2024b) enhances routing accuracy through dual contrastive learning. GraphRouter (Feng et al., 2025) constructs a heterogeneous task-query-LLM graph, formulating model selection as a dynamic edge prediction problem via GNN. Similarly, MODEL-SAT (Zhang et al., 2025b) encodes candidate model performance on core tasks into capability representations, utilizing a lightweight LLM to predict the optimal candidate. Avengers (Zhang et al., 2025a) constructs a training-free router by clustering the question embeddings as the proxies of different question categories, and utilizing the performance of the clusters as a model profile to estimate the LLM candidates.

**Aggregation-based Methods**. Aggregation-based methods typically aggregate the outputs from various LLMs and generate a more accurate and comprehensive response. Agent-Forest (Li et al., 2024a) employs a majority voting mechanism to select the most frequent consensus among responses. LLM-Blender (Jiang et al., 2023) ranks and fuses outputs from diverse LLMs using a pairwise ranker and a generative blender. The Mixture of Agents (MoA) framework (Wang et al., 2024a) introduces a layered architecture that leverages the collective strengths of LLMs through iterative collaboration, significantly improving generation

---

[1]JiSi is named after an idiom that conveys the idea that collective intelligence emerges from aggregating diverse perspectives. The project will be released at https://github.com/magent4aci/openJiSi.

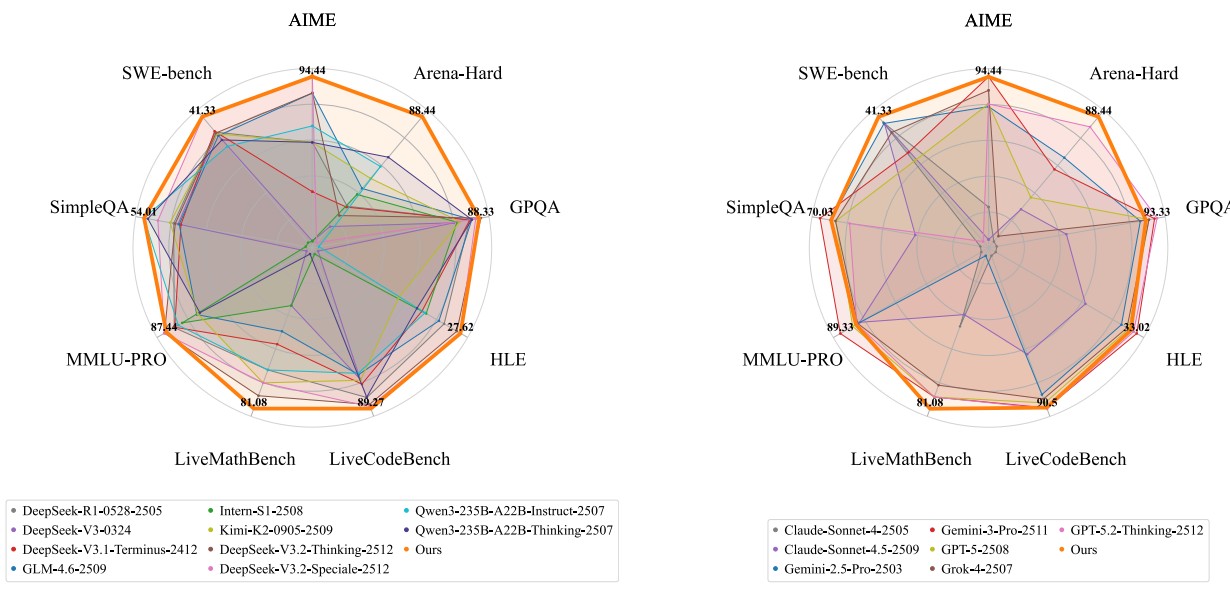

*(a)* Comparison with open-source LLMs.

*(b)* Comparison with closed-source LLMs.

*Figure 2.* Comprehensive performance comparison between the proposed JiSi and other LLMs. (a) and (b) show the fine-grained capability comparison with open-source/closed-source LLMs on different datasets. JiSi achieves superior performance across multiple benchmarks.

quality. Building on this, SMoA (Li et al., 2025a) incorporates response selection and early stopping mechanisms to sparsify information flows among agents, balancing performance with efficiency. Self-MoA (Li et al., 2025b) explores aggregating responses repeatedly sampled from a single high-performance agent. Meanwhile, different from the above ensemble-after-inference methods, ensemble-during-inference methods (Chen et al., 2025b), such as GAC (Yu et al., 2024) and DEEPEN (Huang et al., 2024), aggregate partial inferences of multiple LLMs at the decoding stage and recurrently broadcast the consolidated tokens back to all LLMs.

Besides, recent works have explored unified frameworks that combine routing and aggregation. For instance, Sybomlic-MOE (Chen et al., 2025a) adopts a skill-aware dynamic expert selection strategy, matching tasks to experts based on validation-derived skill keywords and aggregating their outputs. SMACS (Tang et al., 2025) offers a scalable alternative by selecting candidate LLMs through a retrieval-based prior selection and refining outputs via an exploration–exploitation-driven posterior enhancement.

## 3. JiSi Framework

The JiSi framework stems from the rethinking w.r.t. existing routers and aggregation methods. In Fig. 3, we intuitively display the core improvement of JiSi compared with the previous methods in three aspects. In this section, we first introduce the detailed framework of JiSi. Then the rethinking discussion is presented, serving as an analytical summary

of the proposed techniques.

### 3.1. Framework Overview

In summary, JiSi adopts a minimalist "route-and-aggregate" topology: Based on a pre-built embedding bank, it conducts a unified LLM routing and responses aggregation. The processing of a new query comprises three essential components. 1) **Support-Set-based Aggregator Selection**: A support query set is selected for aggregator dispatching and then filtered to improve the assignment of candidate aggregatee models. 2) **Query-Response Mixed Routing**: For each model selected from the filtered support set, a composite prior score is derived from query, response, and reasoning-length similarities, serving as a prior evaluation of the model's expertise for the given query. 3) **Adaptive Routing-Aggregation Switch**: The practical aggregatees are determined based on their prior scores and response length to filter inferior responses and noise. If the number of aggregatees exceeds 1, aggregation is performed to obtain the final response; otherwise, the final response is retrieved from the single aggregatee, *i.e.*, falls back to top-1 routing. An overview of the framework is illustrated in Fig. 4.

### 3.2. Embedding Bank

Considering the inherent heterogeneity and black-box nature of candidate LLMs, the embedding bank is constructed as an LLM profile storage, facilitating architecture-agnostic evaluation. Intuitively, the embedding bank is built on an extensive collection of queries from diverse sources.

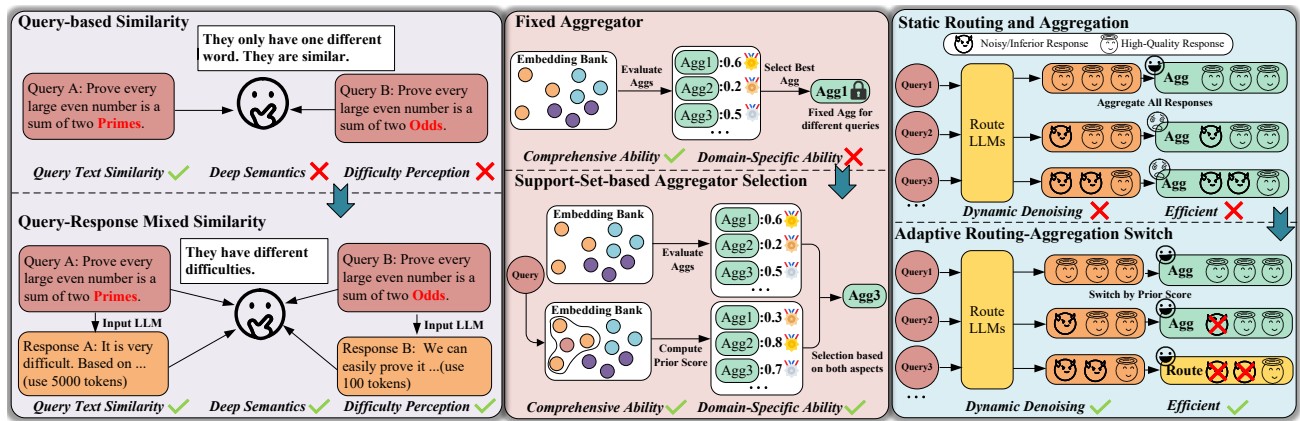

*Figure 3.* JiSi rethinks the existing routing and aggregation methods, and reshapes them from three aspects: 1) **Routing**: from query-based to query-response-mixed; 2) **Aggregation**: from fixed aggregator to support-set-based aggregator selection; 3) **Combination**: from static routing and aggregation to adaptive routing-aggregation switch. For simplicity, the "**Agg**" means Aggregator or Aggregation.

The original queries, LLM responses, and correctness information are subsequently projected into continuous embedding spaces to support similarity analysis. Specifically, let $[U] = [1, U] \cap \mathbb{N}$. Assuming that $N$ questions $\mathcal{Q} = \{x_i \in \mathcal{S} \,|\, i \in [N]\}$ each with corresponding label $y_i \in \mathcal{S}$, and $M$ LLMs $\mathcal{A} = \{\boldsymbol{A}_1, \boldsymbol{A}_2, \ldots, \boldsymbol{A}_M\}$, $\boldsymbol{A}_i : \mathcal{S} \to \mathcal{S}$ are predetermined. The $\mathcal{S}$ contains all literal strings. Responses of each LLM are collected over all questions, obtaining $\mathcal{R} = \{\hat{y}_{i,j} = \boldsymbol{A}_j(x_i) \,|\, i \in [N], j \in [M]\}$ with reasoning costs (completion tokens) $c_{i,j} \in \mathbb{R}_+$. A pre-trained embedding model $\boldsymbol{A}_{emb} : \mathcal{S} \to \mathbb{R}^d$ is introduced to map questions and responses into contiguous latent space, forming the embedding bank $\mathcal{E}$:

$$
\begin{aligned}
\mathcal{E} &= \mathcal{E}^Q \cup \mathcal{E}^R, \\
\mathcal{E}^Q &= \{\boldsymbol{e}_i = \boldsymbol{f}_1(x_i) \,|\, i \in [N]\}, \\
\mathcal{E}^R &= \{\boldsymbol{r}_{i,j} = \boldsymbol{f}_1(\hat{y}_{i,j}) \,|\, i \in [N], j \in [M]\}, \\
\boldsymbol{f}_1(a) &= \frac{\boldsymbol{A}_{emb}(a)}{\|\boldsymbol{A}_{emb}(a)\|_2} \in \mathbb{R}^d, a \in \mathcal{S},
\end{aligned}
\tag{1}
$$

which can be splitted into question embedding set $\mathcal{E}^Q$ and response embedding set $\mathcal{E}^R$. Moreover, for each LLM $\boldsymbol{A}_j$, a capability vector $\boldsymbol{v}_j \in \{0, 1\}^N$ is generated to measure response correctness over questions, *i.e.*, $\boldsymbol{v}_j = \left[\mathbf{1}_{\{y_1\}}(\hat{y}_{1,j}), \ldots, \mathbf{1}_{\{y_N\}}(\hat{y}_{N,j})\right]^T$, where $\mathbf{1}_A(\cdot)$ is a general indicator function, meaning computing the correctness.

### 3.3. Inference

Given a new query, the inference phase is composed of three parts: 1) Support-Set-based Aggregator Selection; 2) Query-Response Mixed Routing; 3) Adaptive Routing-Aggregation Switch. The following sections will introduce this procedure in depth.

**Support-Set-based Aggregator Selection**. Provided a

query denoted as $x$. Its embedding can be extracted as $\boldsymbol{e} = \boldsymbol{f}_1(x)$. Similarities between $\boldsymbol{e}$ and all embeddings in $\mathcal{E}^Q$ are then calculated via cosine distance (note that the embeddings are already normalized): $\boldsymbol{s} = \left[\boldsymbol{e}_1^T \boldsymbol{e}, \ldots, \boldsymbol{e}_N^T \boldsymbol{e}\right]^T \in [0, 1]^N$, where $\boldsymbol{e}_i \in \mathcal{E}^Q$, $i \in [N]$. To determine the support question number $N^{sup}$, a predefined base number $N^{base}$ is used. The indices included in the support set are then derived as $\mathcal{I} = \{i_1, \ldots, i_{N^{sup}}\} = \mathcal{I}\{i \in [N] \,|\, s_i \geq \gamma \left[\boldsymbol{f}_2(N^{base})\right](\boldsymbol{s})\}$, where $\gamma \in [0, 1]$ is predefined and

$$
\begin{aligned}
\boldsymbol{f}_2 : [U] &\to \mathcal{F}, \mathcal{F} = \{\boldsymbol{f} \,|\, \boldsymbol{f} : \mathbb{R}^U \to \mathbb{R}\}, \\
\alpha &\mapsto (\boldsymbol{a} \mapsto \hat{a}), \\
\text{s.t. } |\{i \in [U] \,|\, a_i \geq \hat{a}\}| &= \alpha, \\
\hat{a} \in \{a_i \,|\, i \in [U]\}, U &\in \mathbb{N}_+.
\end{aligned}
\tag{2}
$$

Consequently, $N^{sup}$ can be obtained via $|\mathcal{I}|$. Applying $\mathcal{I}$ to both $\boldsymbol{s}$ and the concatenated $\boldsymbol{v} = \left[\boldsymbol{v}_1^T, \ldots, \boldsymbol{v}_M^T\right]^T \in \{0, 1\}^{M \times N}$ results in $\boldsymbol{s}^{sup} = \boldsymbol{s}[\mathcal{I}] \in [0, 1]^{N^{sup}}$ and $\boldsymbol{v}^{sup} = \boldsymbol{v}[:, \mathcal{I}] \in \{0, 1\}^{M \times N^{sup}}$. The vector slicing operator is denoted as $[\cdot]$. The coarse-grained model prior score $\boldsymbol{g} \in \mathbb{R}_+^M$ is computed via $\boldsymbol{v}^{sup} \boldsymbol{s}^{sup}$. The aggregator $\boldsymbol{A}_{agg} = \boldsymbol{A}_k$ is selected, where $g_k = [\boldsymbol{f}_2(1)](\boldsymbol{g})$. The support set selected provides a broader knowledge base than the subsequent filtered set. The aggregator is anchored in this larger set to ensure a better understanding of divergent domains, thereby enabling more skilled aggregation.

**Query-Response Mixed Routing**. To provide routing priors, the candidate model set $\mathcal{A}^{cm} = \{\boldsymbol{A}_1^{cm}, \ldots, \boldsymbol{A}_K^{cm}\}$ equals to $\{\boldsymbol{A}_i \in \mathcal{A} \,|\, i \in [M], g_i \geq [\boldsymbol{f}_2(K)](\boldsymbol{g})\}$. The given question is then fed into all LLMs in the candidate set, obtaining responses $\hat{y}_k = \boldsymbol{A}_k^{cm}(x)$ and corresponding reasoning costs $c_k \in \mathbb{R}_+$, $k \in [K]$. Each $\hat{y}_k$ will be mapped to its embedding via $\boldsymbol{r}_k = \boldsymbol{f}_1(\hat{y}_k)$. To obtain the fine-grained model prior score, the support set is first filtered, resulting

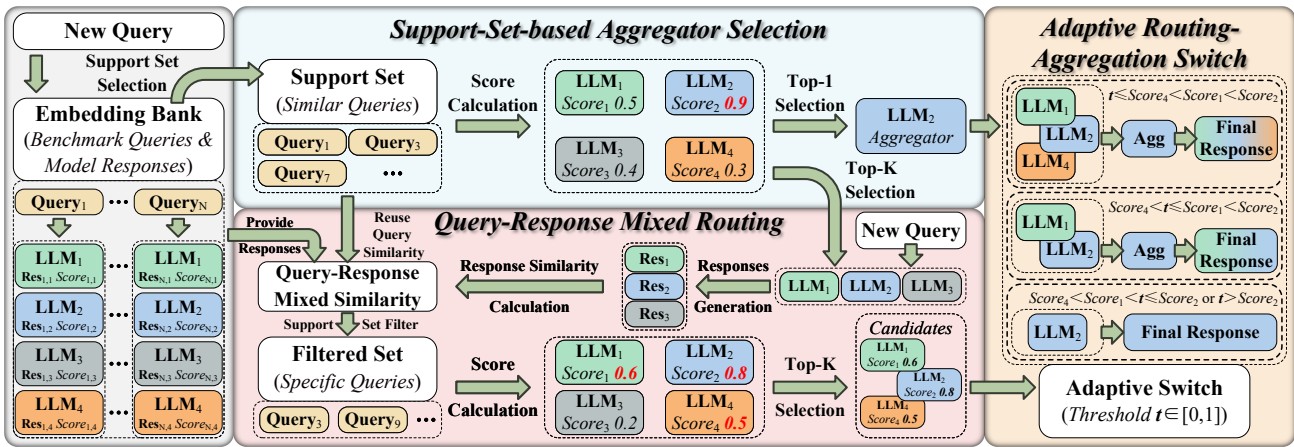

*Figure 4.* Overview of the proposed JiSi. For a new question query, a support set is first extracted from the embedding bank based on question similarity. Consequently, three essential techniques, Support-Set-based Aggregator Selection, Query-Response Mixed Routing, and Adaptive Routing-Aggregation Switch, are combined to select and aggregate LLMs dynamically.

in new indices $\mathcal{I}^{flt}$ containing $N^{flt}$ elements:

$$
\begin{aligned}
\mathcal{I}^{flt} &= \left\{ i \in [N^{sup}] \,\middle|\, s_i^{flt} \geq [\boldsymbol{f}_2(\lfloor \beta N^{sup} \rfloor)]\left(\boldsymbol{s}^{flt}\right) \right\}, \\
\boldsymbol{s}^{flt} &= \left(\epsilon K \boldsymbol{s} + \sigma \boldsymbol{s}^{res} + \delta \boldsymbol{s}^{cost}\right)[\mathcal{I}], \\
s_i^{res} &= \sum\nolimits_{j=1}^{K} \boldsymbol{r}_{i,j}^T \boldsymbol{r}_j, \; \boldsymbol{r}_{i,j} \in \mathcal{E}^R, \; i \in [N], \\
s_i^{cost} &= \sum\nolimits_{j=1}^{K} \left(1 - \left(\left|c_{i,j}^2 - c_j^2\right|/c_{m,j}\right)\right), \; i \in [N], \\
c_{m,j} &= [\boldsymbol{f}_2(1)]\left(\left[\left|c_{i_1,j}^2 - c_j^2\right|, \ldots, \left|c_{i_{N^{sup}},j}^2 - c_j^2\right|\right]\right), \\
&\quad \text{s.t.} \; [\beta, \epsilon, \sigma, \delta] \in [0,1]^4, \; \epsilon + \sigma + \delta = 1.
\end{aligned}
\tag{3}
$$

The hyperparameters $\beta$, $\epsilon$, $\sigma$, $\delta$ are predefined. The filtered indices account for question similarity, model response similarity, and reasoning cost similarity, leading to a more precise support set for better aggregatee selection. With respect to $\mathcal{I}^{flt}$, a fine-grained model prior score is computed as $\boldsymbol{g}^f = \boldsymbol{v}^{sup}\left[:, \mathcal{I}^{flt}\right] \boldsymbol{s}^{flt}\left[\mathcal{I}^{flt}\right] \in \mathbb{R}_+^M$. Compared with the query-only support set, the filtered set accounts for both queries and LLM responses, helping to filter queries that are semantically similar but differ in difficulty, which is captured by the deep semantics and length of LLM responses.

**Adaptive Routing-Aggregation Switch**. To conduct adaptive aggregation, a threshold $t \in [0,1]$ is predetermined. The final aggregatees $\mathcal{A}^{fm}$ are formulated as:

$$
\begin{aligned}
\mathcal{A}^{fm} = \Big\{ &\boldsymbol{A}_i \in \mathcal{A} \,\big|\, i \in [M], \\
&g_i^f \geq [\boldsymbol{f}_2(K)]\left(\boldsymbol{g}^f\right), \left(g_i^f / [\boldsymbol{f}_2(1)]\left(\boldsymbol{g}^f\right)\right) \geq t \Big\}.
\end{aligned}
\tag{4}
$$

Let $N^{fm}$ being $\left|\mathcal{A}^{fm}\right|$ and the elements in $\mathcal{A}^{fm}$ being

$\boldsymbol{A}_i^{fm}$, the final response $\hat{y}$ is gathered via:

$$
\begin{aligned}
\hat{y} &= \begin{cases} \boldsymbol{A}_1^{fm}(x) & \text{if } N^{fm} = 1, \\ \boldsymbol{A}_{agg}(\bar{x}) & \text{if } N^{fm} > 1, \end{cases} \\
\bar{x} &= \boldsymbol{f}_3\left(\left\{\boldsymbol{A}_i^{fm}(x) \,\middle|\, i \in [N^{fm}]\right\}\right).
\end{aligned}
\tag{5}
$$

The $\boldsymbol{f}_3 : 2^{\mathcal{S}} \to \mathcal{S}$ concates any number of strings to a single string. When the number of aggregatees equals 1, aggregation falls back to top-1 fine-grained model prior score routing. Otherwise, the aggregation dynamically selects LLMs (at most K) to provide context. Note that, to obtain $\bar{x}$, the responses $\hat{y}_k$, $k \in [K]$ can be mostly reused due to the overlapping between $\mathcal{A}^{cm}$ and $\mathcal{A}^{fm}$. The adaptive switch fully leverages the advantages of routing and aggregation by filtering noisy LLM responses using a threshold, thereby guaranteeing high-quality aggregation when triggered; otherwise, it provides a more efficient direct-routing path that is less distracting.

### 3.4. Rethinking Routing and Aggregation

**Rethinking Routers**. Current LLM routers use pre-trained embedding models to extract query representations for selecting LLMs (Zhang et al., 2025a; Chen et al., 2024b; Zhuang et al., 2024). While these methods focus on text similarity, they often miss deeper semantics and question difficulty. For instance, the queries "Prove every large even number is a sum of two primes" and "Prove every large even number is a sum of two odds" are similar in wording, but the first is much more difficult. This oversight leads to misrouting of similar queries. To address this, JiSi incorporates latent semantics from generated responses and token costs. Query responses provide deeper insights beyond simple text similarity, enabling effective differentiation based on varying responses. JiSi also factors in token costs, assuming that harder queries incur higher costs, enabling more accu-

rate calibration of query difficulty. By combining query, response, and token costs with appropriate weights, JiSi improves model routing.

**Rethinking Aggregation**. Existing aggregation methods (Li et al., 2025a; Tang et al., 2025) statically select an aggregator based on overall performance, focusing on the comprehensive ability of aggregators, and are inappropriate for a specific task. On the other side, a domain-specific expert does not possess superior aggregation skills. Consequently, the ideal aggregator must strike a balance between domain-specific and comprehensive ability. To identify the optimal task-specific aggregator, JiSi evaluates candidates by considering both their overall performance across broader queries and their specialized expertise within the query's knowledge domain. JiSi utilizes the coarse-grained characteristic of query text similarity. It constructs a larger query support set in a query bank via query-based similarity, which is rich in task information and includes diverse query categories, thereby ensuring balanced estimation of domain-specific expertise and comprehensive ability. In this manner, JiSi can dynamically identify a domain-specific, synthesis-skilled LLM for a new query.

**Rethinking Combination**. Existing methods (Chen et al., 2025a; Tang et al., 2025) typically use a rigid "routing many and aggregating all" approach, treating aggregation as a post-processing step. This overlooks the complementary roles of routing and aggregation; routers are effective for simple tasks while aggregation excels in complex scenarios. Constantly using aggregation can be inefficient due to varying query complexities. Routers select high-quality LLMs, but cannot exceed individual model performance, while aggregation struggles with noisy inputs. To address this, we propose an adaptive routing-aggregation switch mechanism that uses prebuilt embeddings to score candidate responses. This system can prune inferior responses and switch between aggregation and routing. If merely one response exceeds a high-score threshold, it is routed directly, improving inference efficiency.

## 4. Experiment

### 4.1. Settings

**Data Preparation**. To verify the effectiveness of the proposed framework across benchmarks and different LLM compositions, experiments rely heavily on LLMRouter-Bench (Li et al., 2026) (until commit b42d10c) to form both the embedding bank and the test set. For each dataset, the data are split into a training set and a test set in a 0.7:0.3 ratio. *Notably, JiSi is training-free, and the training set is used to construct the embedding bank.* The following benchmarks are involved in the experiments: AIME (MAA, 2024), Arena-Hard (Li et al., 2024b), GPQA (Rein et al.,

2024), HLE (Phan et al., 2025), LiveCodeBench (Jain et al., 2024), LiveMathBench (Liu et al., 2024), MMLU-Pro (Wang et al., 2024b), SimpleQA (Wei et al., 2024), and SWE-bench (Jimenez et al., 2024). For SWE-bench, we pack all context in a single query and drive the LLM or system to solve the issue only via a single response, denoted as SWE-Single-Round. This setting simplifies the evaluation in multi-agent systems and rigorously tests models' tool-free code reasoning and extreme long-context capabilities.

**Implementation Details**. For each training query, the original question and LLM responses are extracted and fed into gte-Qwen2-7B-instruct (Li et al., 2023) to obtain embeddings. Information about completion tokens and response correctness is used to compute reasoning costs and the capability vector. The router and multi-LLM methods are applied to ten open-source LLMs ($M = 10$), including DeepSeek-R1-0528 (DeepSeek-AI et al., 2025a), DeepSeek-V3-0324 (DeepSeek-AI, 2024), DeepSeek-V3.1-Terminus (DeepSeek-AI, 2024), GLM-4.6 (Z.ai, 2025), Intern-S1 (Bai et al., 2025), Kimi-K2-0905 (Team et al., 2025), DeepSeek-V3.2-Thinking (DeepSeek-AI et al., 2025b), DeepSeek-V3.2-Speciale (DeepSeek-AI et al., 2025b), Qwen3-235B-A22B-2507 (Yang et al., 2025), and Qwen3-235B-A22B-Thinking-2507 (Yang et al., 2025). Seven closed-source LLMs are selected for comparison: Claude-Sonnet-4 (Anthropic, 2025a), Claude-Sonnet-4.5 (Anthropic, 2025b), Grok-4 (xAI, 2025), Gemini-2.5-Pro (Comanici et al., 2025), GPT-5 (OpenAI, 2025a), GPT-5.2-Thinking (OpenAI, 2025b) with high thinking, and Gemini-3-Pro (Google, 2025). See Appendix D for details.

### 4.2. Main Results

As demonstrated in Table 1, the proposed JiSi is compared with the aforementioned closed-source and open-source LLMs, along with typical router methods and multi-LLM methods, *i.e.*, RouterDC (Chen et al., 2024b), EmbedLLM (Zhuang et al., 2024), GraphRouter (Feng et al., 2025), Avengers (Zhang et al., 2025a), Retrieval-based Router (Tang et al., 2025), MoA (Wang et al., 2024a), Self Consistency (Chen et al., 2023), Symbolic-MoE (Chen et al., 2025a), Self-MoA (Li et al., 2025b) and Majority Voting (Chen et al., 2024a). For multi-LLM baselines, priors from LLMRouterBench are applied to provide a stronger variant. For each benchmark, the MoA selects three LLMs with the highest evaluation scores over the training records on the benchmark; the Self Consistency sample four rollouts from the best LLM evaluated on training data and merges responses via majority voting; the Majority Voting is verified in two settings. The first is vanilla, encompassing all open-source LLMs, whereas the second comprises only the top-3 LLMs based on the training set. Besides, the best result of open-source LLMs on the test set of each benchmark is gathered to form the model entry "Best LLM" in Table 1.

*Table 1.* Main results over diverse benchmarks, LLMs, and methods. The proposed JiSi exhibits superior performance, ranking the best among all open-source and closed-source LLMs, routing methods, and multi-agent methods. SWE-Single-Round[†]: different from standard SWE-Bench (Jimenez et al., 2024), following the evaluation in (Li et al., 2026), we pack all context in a single query and drive the LLM or system to solve the issue only via a single response to simplify the evaluation in multi-agent systems.

| Model | AIME | Arena-Hard | GPQA | HLE | LiveCodeBench | LiveMathBench | MMLU-Pro | SimpleQA | SWE-Single-Round[†] | Avg |
|---|---|---|---|---|---|---|---|---|---|---|
| *Open-source LLMs* | | | | | | | | | | |
| DeepSeek-R1-0528 (DeepSeek-AI et al., 2025a) | 72.22 | 64.89 | 78.33 | 16.67 | 76.03 | 72.97 | 84.67 | 28.66 | 25.33 | 57.75 |
| DeepSeek-V3-0324 (DeepSeek-AI, 2024) | 38.89 | 59.56 | 68.33 | 3.70 | 61.51 | 59.46 | 78.44 | 26.43 | 24.00 | 46.70 |
| DeepSeek-V3.1-Terminus (DeepSeek-AI, 2024) | 55.56 | 64.67 | 78.33 | 8.64 | 64.67 | 67.57 | 84.56 | 25.12 | 26.00 | 52.79 |
| GLM-4.6 (Z.ai, 2025) | 88.89 | 69.56 | 80.00 | 14.20 | 58.99 | 64.86 | 80.89 | 25.89 | 22.67 | 56.22 |
| Intern-S1 (Bai et al., 2025) | 38.89 | 68.00 | 70.00 | 9.72 | 46.69 | 59.46 | 83.00 | 14.33 | 8.00 | 44.23 |
| Kimi-K2-0905 (Team et al., 2025) | 72.22 | 72.22 | 71.67 | 5.09 | 62.15 | 75.68 | 80.78 | 30.66 | 24.00 | 54.94 |
| DeepSeek-V3.2-Thinking (DeepSeek-AI et al., 2025b) | 88.89 | 62.44 | 88.33 | 24.69 | 83.91 | 78.38 | 87.33 | 27.81 | 24.67 | 62.94 |
| **DeepSeek-V3.2-Speciale** (DeepSeek-AI et al., 2025b) | 94.44 | 55.33 | 83.33 | 27.16 | 86.75 | 75.68 | 87.44 | 39.52 | 40.67 | **65.59** |
| Qwen3-235B-A22B-2507 (Yang et al., 2025) | 77.78 | 75.33 | 80.00 | 9.41 | 58.36 | 72.97 | 83.78 | 54.01 | 16.67 | 55.92 |
| Qwen3-235B-A22B-Thinking-2507 (Yang et al., 2025) | 72.22 | 77.78 | 80.00 | 7.56 | 75.71 | 48.65 | 80.56 | 49.31 | 20.00 | 56.87 |
| *Closed-source LLMs* | | | | | | | | | | |
| Claude-Sonnet-4 (Anthropic, 2025a) | 41.11 | 55.47 | 71.33 | 4.60 | 56.85 | 62.16 | 83.58 | 15.58 | 35.33 | 47.33 |
| Claude-Sonnet-4.5 (Anthropic, 2025b) | 27.78 | 64.00 | 71.67 | 7.56 | 60.57 | 59.46 | 86.33 | 16.18 | 34.00 | 47.51 |
| Gemini-2.5-Pro (Comanici et al., 2025) | 82.22 | 77.64 | 84.00 | 20.62 | 77.54 | 45.95 | 86.69 | 54.33 | 34.53 | 62.61 |
| Grok-4 (xAI, 2025) | 88.89 | 56.89 | 88.33 | 24.42 | 81.03 | 75.68 | 86.56 | 48.38 | 27.33 | 64.17 |
| GPT-5 (OpenAI, 2025a) | 83.33 | 67.11 | 88.33 | 25.77 | 84.54 | 78.38 | 87.22 | 48.00 | 16.00 | 64.30 |
| GPT-5.2-Thinking (OpenAI, 2025b) (High Effort) | 83.33 | 85.78 | 93.33 | 29.94 | 90.50 | 78.38 | 86.67 | 35.21 | 12.67 | 66.20 |
| **Gemini-3-Pro** (Google, 2025) | 94.44 | 74.55 | 91.67 | 33.02 | 89.59 | 78.38 | 89.33 | 70.03 | 18.00 | **71.00** |
| *Router Methods* | | | | | | | | | | |
| RouterDC (Chen et al., 2024b) | 88.89 | 76.00 | 73.33 | 21.30 | 83.60 | 78.38 | 84.33 | 53.31 | 24.00 | 64.79 |
| EmbedLLM (Zhuang et al., 2024) | 94.44 | 57.78 | 90.00 | 27.31 | 84.86 | 75.68 | 87.44 | 53.70 | 42.67 | 68.21 |
| GraphRouter (Feng et al., 2025) (Performance-only) | 88.89 | 82.67 | 90.00 | 25.93 | 86.12 | 78.38 | 87.33 | 49.08 | 27.33 | 68.41 |
| Avengers (Zhang et al., 2025a) | 94.44 | 70.44 | 83.33 | 27.16 | 85.49 | 75.68 | 87.33 | 54.08 | 40.67 | 68.74 |
| Retrieval-based Router (Tang et al., 2025) | 94.44 | 72.67 | 86.67 | 26.85 | 85.17 | 75.68 | 86.78 | 53.78 | 40.00 | 69.12 |
| **Our Query-Response Mixed Router** | 94.44 | 76.89 | 83.33 | 26.70 | 85.17 | 75.68 | 87.00 | 54.62 | 43.33 | **69.68** |
| *Multi-LLM/Multi-Response Methods* | | | | | | | | | | |
| Best (open-source) LLM | 94.44 | 77.78 | 88.33 | 27.16 | 86.75 | 78.38 | 87.44 | 54.01 | 40.67 | 70.55 |
| MOA (Wang et al., 2024a) (3 Best LLMs) | 94.44 | 80.00 | 78.33 | 28.70 | 88.33 | 75.68 | 86.44 | 42.99 | 33.33 | 67.58 |
| Self Consistency (Chen et al., 2023) (Best LLM with 4 Rollouts) | 94.44 | 83.70 | 85.00 | 27.62 | 88.96 | 75.68 | 87.56 | 54.31 | 38.67 | 70.66 |
| Majority Voting (Chen et al., 2024a) (10 LLMs) | 83.33 | 74.67 | 85.00 | 13.58 | 72.56 | 72.97 | 87.44 | 36.98 | 28.67 | 61.69 |
| Majority Voting (Chen et al., 2024a) (3 Best LLMs) | 88.89 | 72.44 | 86.67 | 25.93 | 85.80 | 75.68 | 86.78 | 30.20 | 29.33 | 64.64 |
| Symbolic-MoE (Chen et al., 2025a) | 94.44 | 73.11 | 83.33 | 27.47 | 88.96 | 75.68 | 87.00 | 52.70 | 19.33 | 66.89 |
| Self-MoA (Li et al., 2025b) | 88.89 | 80.44 | 86.67 | 30.09 | 84.54 | 75.68 | 87.00 | 55.39 | 10.00 | 66.52 |
| **JiSi (Ours)** | 94.44 | 88.44 | 86.67 | 27.62 | 89.27 | 81.08 | 86.78 | 53.70 | 41.33 | **72.15** |

**Superiority of JiSi**. The proposed JiSi framework exhibits promising performance, ranking first on average among all closed-source LLMs, open-source LLMs, router methods, and multi-LLM methods. The performance gain over the most powerful open-source LLM (DeepSeek-V3.2-Speciale) is substantially 6.56%. Even compared with the state-of-the-art closed-source LLM Gemini-3-Pro, a +1.15% gain is achieved. Moreover, the router-only variant of JiSi, which directly adopts $A_1^{fm}$ without aggregation formulated in Eq. 5, outperforms other router methods, indicating that the proposed query-response mixed router can yield more suitable aggregatees for subsequent aggregation. Compared with the router-only variant, JiSi surpasses the best method by +3.03%. Besides, JiSi overpasses the strong performance bound of routing, *i.e.* "Best LLM", by +1.6%, revealing enormous potential. Meanwhile, we also conduct the ablation study of JiSi in Appendix C.

**Efficiency of JiSi.** As shown in Table 2, we report the cost of the proposed JiSi and the leading proprietary LLMs. It is worth noting that although JiSi utilizes more API calls due to multi-LLM aggregation, it can remarkably reduce the total cost while maintaining superior performance compared with leading proprietary LLMs. For instance, JiSi surpasses Gemini-3-Pro and Grok-4 by +1.15% and +7.89%, while saving 53.23% and 81.68% cost. The reasons lie in two aspects: 1) the APIs of open-source LLMs are dramatically cheaper than proprietary LLMs. 2) The adaptive routing-

aggregation switch mitigates the workload of aggregation and even simplifies aggregation to routing. The strong trade-off between efficiency and effectiveness demonstrates the feasibility and economy of JiSi in practical implementation. See Appendix D.2 for the detailed LLM prices.

### 4.3. Scaling Ability

To verify the scalability of JiSi, we initialize a base LLM bank with five LLMs: DeepSeek-V3-0324, DeepSeek-R1-0528, Intern-S1, Qwen3-235B-A22B-2507, and Qwen3-235B-A22B-Thinking-2507. Subsequently, we introduce five additional open-source LLMs following their chronological release sequence: Kimi-K2-0905, DeepSeek-V3.1-Terminus, GLM-4.6, DeepSeek-V3.2-Thinking, and DeepSeek-V3.2-Speciale. This setup allows us to simulate the real-world evolution of the LLM ecosystem, resulting in five experimental configurations with model pool sizes $\{5, 6, 7, 8, 9, 10\}$. All other hyperparameters remain the same as Sec. 4.1. To provide a multi-faceted evaluation, we categorize nine benchmarks into three specialized domains: 1) **Reasoning & Knowledge**: AIME, GPQA, MMLU-Pro, HLE, LiveMathBench. 2) **Coding & Engineering**: LiveCodeBench, SWE-bench. 3) **General Chat & Factuality**: Arena-Hard, SimpleQA. As illustrated in Fig. 5, JiSi demonstrates a robust and consistent performance gain across all three domains as the number of candidate LLMs increases.

*Table 2.* The cost report of the proposed open-source collective JiSi and leading proprietary LLMs. The cost is measured by U.S. dollars. Compared with proprietary LLMs, JiSi can save approximately 40% ∼ 80% cost with better performance.

| | AIME | Arena-Hard | GPQA | HLE | LiveCodeBench | LiveMathBench | MMLU-Pro | SimpleQA | SWE-bench | All Cost (JiSi saving ratio) | Avg Performance |
|---|---|---|---|---|---|---|---|---|---|---|---|
| Grok-4 (xAI, 2025) | $3.98 | $11.11 | $8.97 | $171.56 | $45.78 | $2.69 | $50.08 | $22.44 | $29.24 | $345.85 (↓ 81.68%) | 64.17 |
| GPT-5 (OpenAI, 2025a) | $1.57 | $7.97 | $2.22 | $39.29 | $18.17 | $1.43 | $10.25 | $16.98 | $7.49 | $105.37 (↓ 39.87%) | 64.30 |
| GPT-5.2-Thinking (OpenAI, 2025b) | $1.07 | $5.39 | $1.64 | $52.49 | $15.62 | $1.18 | **$6.49** | $16.80 | $11.52 | $112.20 (↓ 43.53%) | 66.20 |
| Gemini-3-Pro (Google, 2025) | $1.73 | $4.57 | $3.18 | $52.47 | $22.91 | $1.97 | $23.29 | $7.56 | $17.78 | $135.46 (↓ 53.23%) | 71.00 |
| JiSi (Open-source Collective) | **$0.61** | **$2.95** | **$1.23** | **$31.01** | **$9.38** | **$0.59** | $7.12 | **$4.85** | **$5.62** | **$63.36** | 72.15 |

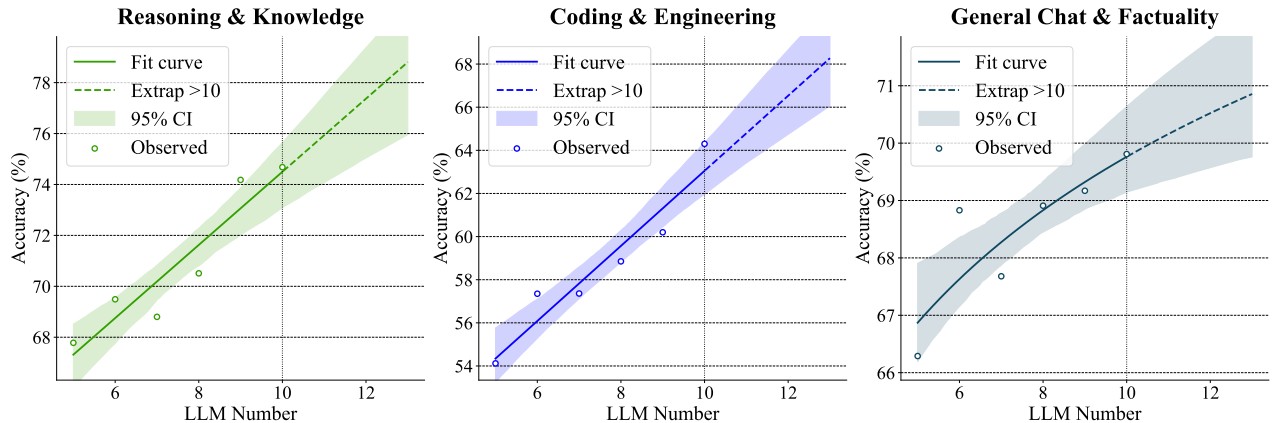

*Figure 5.* The scaling curve of the proposed JiSi, where we add open-source LLMs following their chronological release sequence as a simulation of the real-world evolution of the LLM ecosystem.

The consistent improvement across diverse benchmarks validates that JiSi is not merely a static ensemble but a dynamic system that scales with the progress of the open-source community. This empirical evidence supports a paradigm shift in intelligence enhancement, transitioning from the neural scaling law of parameters and training data towards a novel scaling law of multi-LLM collaboration. Moreover, we also demonstrate the task scaling ability of JiSi in Apendix B.

### 4.4. Analysis on Similarity Evaluation

To evaluate the effectiveness of the proposed query-response mixed similarity discussed in Sec. 3, we propose a new metric. For a given query $x$, the support set can be constructed using various similarity criteria: query similarity only, query with response similarity, and the method adopted by JiSi, which incorporates query, response, and length (token costs) similarity. Once the support set is obtained, a fine-grained model score $g$ can be computed in accordance with Eq. 3. This predicted model score can then be compared to the actual label $v$ using the formula $100 \, |g - v|_1 \, /M$. This represents the deviation between the optimal support set and the actual selected set, and the less the better. In Table 3, this metric is summarized for each benchmark based on the three similarity criteria mentioned earlier. It is evident that, by adding response and length similarity, the deviations are reduced by 1.44% and 1.69%, respectively. This indicates that the proposed mixed-similarity approach provides a more accurate estimate of model performance, causing more reliable routing results for subsequent aggregation.

### 4.5. Reference Accuracy Study

The proposed Adaptive Routing-Aggregation Switch is designed to filter out ineffective aggregatees while retaining valuable ones. In Table 4, we present the effects of this switching mechanism. To perform a comprehensive evaluation, the embedding bank is divided into three categories based on question difficulty. The detailed division metric is shown in Appendix D.5. The results indicate that the switching mechanism consistently improves aggregatee quality by eliminating less effective ones, particularly for "Hard" questions, where we observe an accuracy increase of 8.49%. This improvement in aggregator performance ensures reliable aggregation by yielding more correct responses.

### 4.6. Aggregator Selection Study

To validate the aggregator selection rethinking in Sec. 3, we fix three reference responses for each test query while varying the size of the support set to select the aggregator. As the support set size increases, the aggregator exhibits diminished domain-specific capability but enhanced comprehensive capability. As shown in Fig. 6, the aggregator yields suboptimal performance when the aggregator leans excessively towards either domain-specificity or comprehensiveness. The optimal performance is achieved when these two capabilities are well-balanced. It reveals the mechanism behind the proposed Support-Set-based Aggregator Selection, which aims to strike a balance between two capabilities.

*Table 3.* The deviation of model performance estimation via different similarity criteria. ↓: The less is the better.

|  | AIME ↓ | Arena-Hard ↓ | GPQA ↓ | HLE ↓ | LiveCodeBench ↓ | LiveMathBench ↓ | MMLU-Pro ↓ | SimpleQA ↓ | SWE-bench ↓ | Avg ↓ |
|---|---|---|---|---|---|---|---|---|---|---|
| Query | 36.42 | 35.96 | 35.18 | 22.33 | 37.93 | 39.72 | 30.01 | 37.76 | 34.31 | 34.40 |
| Query+Response | 35.26 | 34.10 | 33.43 | 22.04 | 33.29 | 38.85 | 27.84 | 37.00 | 34.80 | 32.96 |
| Query+Response+Length | 30.92 | 33.91 | 31.11 | 22.51 | 28.30 | 37.60 | 25.40 | 37.11 | 34.53 | 31.27 |

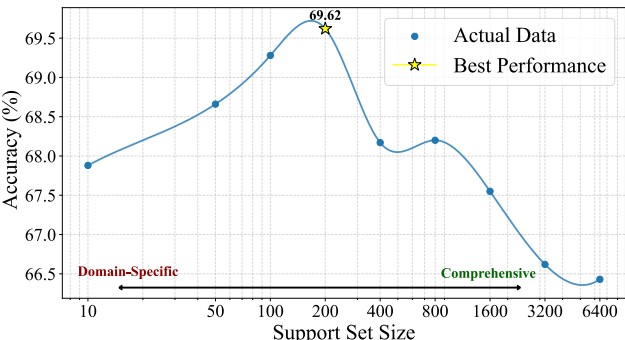

*Figure 6.* The performance curve of aggregators with different domain-specific and comprehensive abilities.

*Table 4.* The accuracy(%) of 1) all 2) removed 3) remained reference responses when applying adaptive switch.

|  | All Ref Acc | Removed Ref Acc | Remained Ref Acc |
|---|---|---|---|
| Easy | 95.37 | 93.95(-1.42) | 98.07(+2.7) |
| Medium | 71.97 | 67.54(-4.43) | 77.11(+5.14) |
| Hard | 22.25 | 16.46(-5.79) | 30.74(+8.49) |

## 5. Conclusion

We propose JiSi, a training-free and scalable multi-LLM system that leverages a minimalist "Route-and-Aggregate" topology. By incorporating three innovations, including query-response mixed routing, support-set-based aggregator selection, and adaptive routing-aggregation switch, JiSi orchestrates the collaboration of open-source LLMs while mitigating noise. Experiments across nine benchmarks demonstrate that JiSi not only significantly outperforms existing multi-LLM baselines but also surpasses the leading proprietary model, Gemini-3-Pro, while saving 53.23% cost. It verifies that the strategic collaboration of models provides an effective and scalable alternative to the continuous expansion of monolithic models.

## Acknowledgements

This work was supported by New Generation Artificial Intelligence-National Science and Technology Major Project (2025ZD0121801) and the JC STEM Lab of AI for Science and Engineering, funded by The Hong Kong Jockey Club Charities Trust, the Research Grants Council of Hong Kong (Project No. CUHK14213224).

## Impact Statement

This paper presents work whose goal is to advance the field of Machine Learning. There are many potential societal consequences of our work, none which we feel must be specifically highlighted here.

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

## A. Distribution of LLM Selection

To further investigate the underlying routing logic of JiSi, we visualize the usage distribution of selected models across nine benchmarks in Fig. 7. The empirical distribution reveals three key insights.

**Preference for High-Capacity LLMs.** As illustrated in the overall distribution, JiSi performs a distinct preference for high-capacity models such as DeepSeek-V3.2-Speciale and Qwen3-235B-A22B-2507. By mainly routing queries to these top-performing open-source models with the highest individual average scores shown in Table 1, the system establishes a superior intelligence floor. This strategic selection ensures that the subsequent aggregation process operates on high-fidelity information, providing the necessary reasoning density for final high aggregation performance.

**Task-Aware Specialization and Domain Expertise.** JiSi exhibits remarkable sensitivity to the inherent requirements of different domains, effectively matching tasks with specialized experts. For instance, in reasoning-heavy tasks such as AIME and HLE, the router converged almost exclusively on DeepSeek-V3.2-Speciale. Meanwhile, for SimpleQA, which demands vast world knowledge and factuality, the system dynamically shifted its preference towards Qwen3-235B-A22B, which is the model with the fewest hallucinations and highest factual accuracy. It demonstrates that JiSi does not merely select the best overall model, but identifies the optimal matching between given query semantics and model expertise.

**Abundant Usage Diversity against System Collapse.** Crucially, JiSi avoids the pitfall of system collapse, where the system redundantly relies on a single LLM and degrades into a single model. This is most evident in Arena-Hard and MMLU-Pro, where the selection entropy is high, involving a diverse ensemble including DeepSeek-R1-0528, Kimi-K2, and Intern-S1. This heterogeneity is vital, which guarantees that the aggregator receives complementary perspectives, forming a prerequisite for the emergent performance gains. The preservation of this diverse candidate pool mitigates single-point biases and enhances the system's generalization across multifaceted user prompts.

## B. Task Scaling

To demonstrate JiSi's ability to handle increasingly added tasks, we conduct experiments on scientific research scenarios with SGIbench (Xu et al., 2025). The new scientific tasks include 1) Wet Experiment 2) Dry Experiment 3) Deep Research 4) Idea Generation, which are different from the existing tasks in Sec. 4. For the task extension setting, we maintain the original embedding bank, and follow the data preparation in Sec. 4.1 to add new task-corresponding queries and responses. The results are shown in Table 6. The proposed JiSi effectively handles novel tasks, outperforming the best open-source LLM by 3.53%, achieving an average improvement of 1.76% over the leading closed-source LLM. It demonstrates JiSi's immense potential for task scaling. It can be easily and efficiently adapted to newly added, distinct tasks, verifying the feasibility of JiSi in practical implementation.

## C. Ablation Study

To evaluate the impact of each proposed component, detailed ablation studies are presented in Table 5. For brevity, we refer to the ablated components by the following abbreviations: QRR (Query-Response Mixed Routing), SAS (Support-Set-based Aggregator Selection), and ARS (Adaptive Routing-Aggregation Switch). As components are gradually removed, a consistent decline in performance is observed: a 1.06% drop after removing ARS, an additional 1.41% decrease after further eliminating SAS, and ultimately, a total accuracy loss of 4.32% when no components are enabled. The results demonstrate that orchestrating all components contributes to JiSi's extraordinary performance. Besides, it is worth noting that simply adopting ARS negatively affects performance, meaning that without the guidance of QRR and SAS, ARS lacks a valuable metric to dispatch routing and aggregation, further demonstrating the importance of composing all components together.

## D. Implementation Details

### D.1. Dataset Details

We follow most of the dataset settings in LLMRouterBench (Li et al., 2026). Specifically, we select the nine most challenging datasets in LLMRouterBench to evaluate the proposed JiSi and other leading LLMs. For each dataset, the data are split into a training set and a test set in a 0.7:0.3 ratio. All data are first added to a list in ascending order by the recorded index, and then shuffled using Numpy (Harris et al., 2020) with seed 42. The training set comprises the first 0.7 proportion of the shuffled list, and the test set comprises the remainder. If a response is missing during embedding bank construction or test set inference,

an API remote call is triggered to retrieve it from LLM providers, given temperature 0.2 and top-p 1.0. The received response is then cached back to LLMRouterBench. The evaluation also follows (Li et al., 2026) and uses non-interactive mode with a single-round response. Specifically, the judger for Arena-Hard is DeekSeek-V3-0324 (DeepSeek-AI, 2024), and the judger for HLE and SimpleQA is o3-mini (OpenAI, 2025c). For SWE-bench, because aggregation methods are hard to conduct multi-turn interaction and tool usage, we pack all context in a single query and drive the LLM/system to solve the issue only via a single response following (Li et al., 2026). For an LLM, its responses on the SWE-bench test set are packaged and sent to sb-cli to finalize the results. Besides, we introduce SGIBench (Xu et al., 2025) as an additional distinct dataset to verify the task scaling ability of JiSi. The detailed descriptions of all used datasets are as follows:

**AIME.** This benchmark features challenging numeric-answer problems inspired by the American Invitational Mathematics Examination. It encompasses topics such as algebra, geometry, number theory, and combinatorics, specifically designed to test and enhance the reasoning abilities required for high-school level math competitions.

**Arena-Hard.** Arena-Hard is a 500-prompt, open-ended benchmark built by the BenchBuilder pipeline, which automatically filters and samples diverse, high-quality prompts from large crowdsourced datasets like Chatbot Arena and WildChat-1M. It evaluates models via LLM-as-a-judge pairwise comparisons, yielding strong model separability and very high alignment with human preferences, while costing about $20 per model and supporting frequent, human-free updates.

**GPQA.** The GPQA benchmark pushes for deep, conceptual understanding in graduate-level physics. It covers a wide range of topics, including classical mechanics, electromagnetism, quantum mechanics, thermodynamics, and more, requiring advanced reasoning to tackle complex questions.

**HLE.** A multi-modal benchmark designed to be the final closed-ended test for AI, consisting of 2,500 questions across mathematics, humanities, and the natural sciences. To address the saturation of earlier benchmarks like MMLU, this dataset was constructed by experts with a strict filtering process: questions were rejected if current frontier models could already solve them. As a result, state-of-the-art LLMs demonstrate low accuracy on HLE, highlighting the remaining gap between AI capabilities and the frontier of expert human knowledge.

**LiveCodeBench.** This benchmark offers a comprehensive, contamination-free evaluation framework for code-centric LLMs. It sources dynamic problems from contest platforms like LeetCode, AtCoder, and CodeForces to mitigate data contamination. Beyond code generation, it specifically targets broader capabilities such as self-repair, code execution, and test output prediction to assess holistic programming proficiency.

**LiveMathBench.** Featuring 140 recently curated problems from prestigious mathematics competitions such as the CNMO, CCEE, AMC, and Putnam, LiveMathBench 2024 tests the stability of mathematical reasoning in LLMs.

**MMLU-Pro.** Building upon the MMLU benchmark, MMLU-Pro introduces 1000 additional, more challenging questions across 14 disciplines, including new tasks with detailed chain-of-thought annotations. This enhanced version provides a rigorous test of university-level multi-task reasoning and comprehension across diverse academic fields.

**SimpleQA.** This benchmark evaluates short-form factuality by testing whether language models can answer concise, unambiguous, fact-seeking questions. Collected adversarially against GPT-4 and constructed so that each question has a single indisputable answer, it grades responses as correct, incorrect, or not attempted, encouraging models to answer only when confident.

**SWE-bench.** SWE-bench is a realistic benchmark for evaluating language models on end-to-end software engineering: given a full Python codebase and a real GitHub issue, models must produce a patch (a pull request) that fixes the bug or implements the feature and passes the repository's fail-to-pass tests. Built from thousands of issues across 12 projects, it stresses long-context understanding, cross-file edits, and execution-verified solutions, revealing that current models solve only a small fraction of cases while providing a scalable testbed for progress.

**SGIBench.** This benchmark evaluates Scientific General Intelligence (SGI) by testing whether LLMs can autonomously navigate the full, iterative cycle of scientific inquiry: Deliberation, Conception, Action, and Perception. It consists of over 1,000 expert-curated samples across 10 disciplines, inspired by Science's 125 Big Questions. The benchmark is structured into five scientist-aligned task families, including deep research, idea generation, dry experiments, wet experiments, and multimodal experimental reasoning. By filtering for high-difficulty tasks, it exposes critical bottlenecks in current models regarding quantitative synthesis, experimental trajectory planning, and complex multimodal discrimination. Because multimodal settings exceed the scope of this paper, the multimodal experimental reasoning subset is not tested.

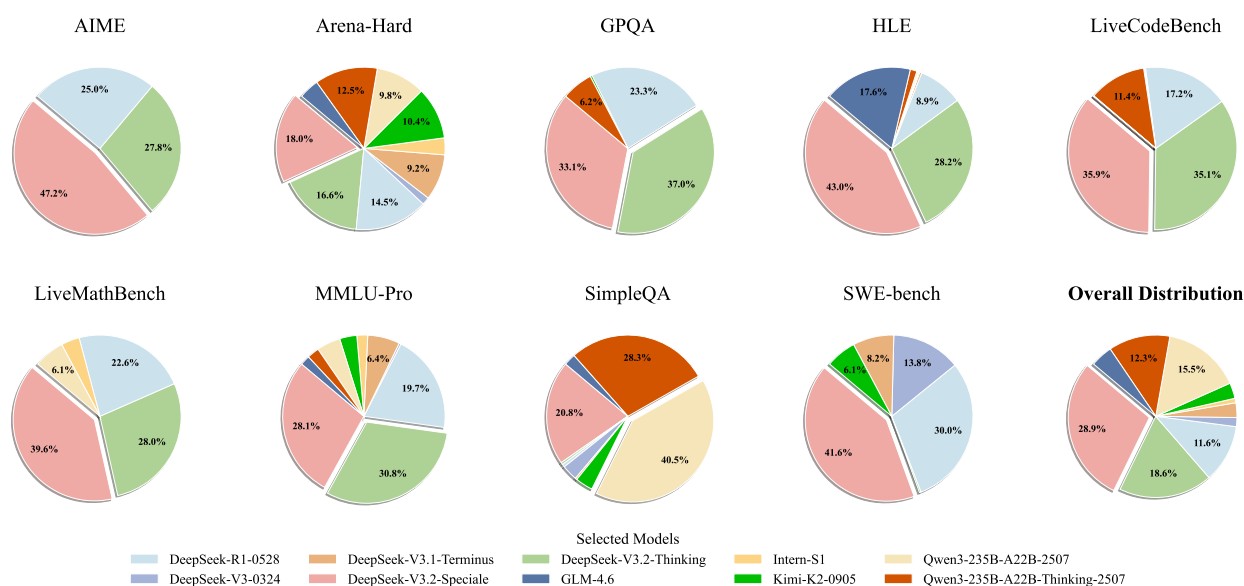

*Figure 7.* The distribution of LLM selection using the proposed JiSi on different datasets. JiSi prefers to select the high-performance LLMs while maintaining the task-aware specialization and abundant usage diversity.

*Table 5.* The component ablation study. QRR: Query-Response Mixed Routing; SAS: Support-Set-based Aggregator Selection; ARS: Adaptive Routing-Aggregation Switch.

| QRR | SAS | ARS | Reasoning & Knowledge | Coding & Engineering | General Chat & Factuality | Avg |
|---|---|---|---|---|---|---|
| ✓ | ✓ | ✓ | 75.32 | 65.30 | 71.07 | 72.15 |
| ✓ | ✓ | ✗ | 75.07 | 63.30 | 68.95 | 71.09 |
| ✓ | ✗ | ✗ | 73.43 | 64.25 | 65.76 | 69.68 |
| ✗ | ✗ | ✗ | 72.28 | 58.85 | 65.70 | 67.83 |
| ✗ | ✗ | ✓ | 67.04 | 56.20 | 68.13 | 64.87 |
| ✗ | ✓ | ✗ | 74.11 | 57.75 | 66.32 | 68.74 |
| ✗ | ✓ | ✓ | 73.86 | 59.67 | 66.80 | 69.14 |

*Table 6.* Task scaling study on four distinct scientific tasks. JiSi can be easily scaled on other tasks by increasingly extending the embedding bank and exhibits strong performance.

| | Wet Experiment | Dry Experiment | Deep Research | Idea Generation | Avg |
|---|---|---|---|---|---|
| DeepSeek-R1-0528 (DeepSeek-AI et al., 2025a) | 22.18 | 29.27 | 13.54 | 25.31 | 22.58 |
| DeepSeek-V3-0324 (DeepSeek-AI, 2024) | 29.88 | 23.17 | 14.58 | 24.24 | 22.97 |
| DeepSeek-V3.1-Terminus (DeepSeek-AI, 2024) | 15.97 | 20.73 | 10.42 | 26.87 | 18.50 |
| GLM-4.6 (Z.ai, 2025) | 24.27 | 29.27 | 13.54 | 24.81 | 22.97 |
| Intern-S1 (Bai et al., 2025) | 30.81 | 26.83 | 12.50 | 24.42 | 23.64 |
| Kimi-K2-0905 (Team et al., 2025) | 22.81 | 25.61 | 12.50 | 35.98 | 24.23 |
| DeepSeek-V3.2-Thinking (DeepSeek-AI et al., 2025b) | 16.91 | 24.39 | 11.46 | 17.22 | 17.50 |
| DeepSeek-V3.2-Speciale (DeepSeek-AI et al., 2025b) | 16.91 | 29.27 | 14.58 | 26.18 | 21.74 |
| Qwen3-235B-A22B-2507 (Yang et al., 2025) | 27.52 | 19.51 | 11.46 | 25.57 | 21.02 |
| Qwen3-235B-A22B-Thinking-2507 (Yang et al., 2025) | 17.20 | 25.61 | 13.54 | 16.51 | 18.22 |
| Claude-Sonnet-4 (Anthropic, 2025a) | 28.76 | 26.83 | 10.42 | 28.94 | 23.74 |
| Claude-Sonnet-4.5 (Anthropic, 2025b) | 20.02 | 30.49 | 12.50 | 34.62 | 24.41 |
| Gemini-2.5-Pro (Comanici et al., 2025) | 19.79 | 32.93 | 14.58 | 27.32 | 23.66 |
| Gemini-3-Pro (Google, 2025) | 30.71 | 34.15 | 13.54 | 25.58 | 26.00 |
| GPT-5 (OpenAI, 2025a) | 8.37 | 26.83 | 13.54 | 45.68 | 23.61 |
| GPT-5.2-Thinking (OpenAI, 2025b) (High Effort) | 9.30 | 29.27 | 16.67 | 42.93 | 24.54 |
| JiSi (Ours) | 29.20 | 30.49 | 15.63 | 35.71 | **27.76** |

*Table 7.* The detailed parameters, release dates, and API prices of the corresponding LLMs.

| | Open-Source | Parameters | Input Price ($/M tokens) | Output Price ($/M tokens) | Release Date |
|---|---|---|---|---|---|
| DeepSeek-R1-0528 (DeepSeek-AI et al., 2025a) | ✓ | 685B | 0.40 | 1.75 | 2025-05-28 |
| DeepSeek-V3-0324 (DeepSeek-AI, 2024) | ✓ | 685B | 0.19 | 0.87 | 2025-03-24 |
| DeepSeek-V3.1-Terminus (DeepSeek-AI, 2024) | ✓ | 685B | 0.21 | 0.79 | 2025-09-22 |
| GLM-4.6 (Z.ai, 2025) | ✓ | 357B | 0.42 | 1.97 | 2025-09-30 |
| Intern-S1 (Bai et al., 2025) | ✓ | 241B | $\approx 0.28$ | $\approx 0.42$ | 2025-07-26 |
| Kimi-K2-0905 (Team et al., 2025) | ✓ | 1000B | 0.63 | 2.64 | 2025-09-05 |
| DeepSeek-V3.2-Thinking (DeepSeek-AI et al., 2025b) | ✓ | 685B | 0.28 | 0.42 | 2025-12-01 |
| DeepSeek-V3.2-Speciale (DeepSeek-AI et al., 2025b) | ✓ | 685B | 0.28 | 0.42 | 2025-12-15 |
| Qwen-235B-A22B-2507 (Yang et al., 2025) | ✓ | 235B | 0.28 | 1.13 | 2025-07-25 |
| Qwen3-235B-A22B-Thinking-2507 (Yang et al., 2025) | ✓ | 235B | 0.35 | 1.41 | 2025-07-25 |
| Claude-Sonnet-4 (Anthropic, 2025a) | ✗ | ? | 3.00 | 15.00 | 2025-05-23 |
| Claude-Sonnet-4.5 (Anthropic, 2025b) | ✗ | ? | 3.00 | 15.00 | 2025-09-29 |
| Gemini-2.5-Pro (Comanici et al., 2025) | ✗ | ? | 1.25 | 10.00 | 2025-05-06 |
| Grok-4 (xAI, 2025) | ✗ | ? | 3.00 | 15.00 | 2025-04-01 |
| GPT-5 (OpenAI, 2025a) | ✗ | ? | 1.25 | 10.00 | 2025-08-07 |
| GPT-5.2-Thinking (OpenAI, 2025b) | ✗ | ? | 1.13 | 9.03 | 2025-12-11 |
| Gemini-3-Pro (Google, 2025) | ✗ | ? | 1.05 | 6.31 | 2025-11-19 |

### D.2. LLM Details

Our LLM bank is curated with a cutoff date of January 2026. To fully leverage the potential of JiSi, we selected ten of the most prevalent and powerful open-source models to construct our LLM bank, spanning five independent organizations to ensure architectural and institutional diversity. Furthermore, we evaluated JiSi against seven leading proprietary LLMs from four different providers. Details, including parameter counts, pricing, and release dates, are summarized in Table 7.

### D.3. Details of Compared Methods

**GraphRouter.** Our implementation leverages the official GraphRouter (Feng et al., 2025) framework. We generate embeddings for queries, tasks, and model descriptions using gte-Qwen2-7B-instruct (Li et al., 2023), with input layer dimensions adjusted accordingly. To address potential label bias arising from tied predictions, we substitute standard argmax-based one-hot labels with multi-hot supervision, encompassing all tie-optimal models for each query. To ensure robust training stability, the model—which comprises approximately 0.1M trainable parameters—is trained over 10,000 epochs. Following the original study, we evaluate the model across three standard configurations: Performance First (PF), Balance (BL), and Cost First (CF).

**RouterDC.** We adapt the official RouterDC (Chen et al., 2024b) framework, substituting the default encoder with gte-Qwen2-7B-instruct (Li et al., 2023) to ensure a fair comparison via a unified embedding backbone. The resulting 7B-parameter model is trained using DeepSpeed across eight NVIDIA A800-80G GPUs, with a per-device batch size of 8. To maintain consistency, all other hyperparameters are kept identical to the original configuration.

**EmbedLLM.** We build upon the official EmbedLLM (Zhuang et al., 2024) framework, integrating gte-Qwen2-7B-instruct (Li et al., 2023) for query embedding generation to ensure cross-method consistency. The input layer is reconfigured to accommodate the resulting embedding dimensions. To enhance training stability, we scale the batch size to 32,768, while maintaining all other hyperparameters in alignment with the original study. The final model comprises approximately 12M trainable parameters.

**Avengers.** We employ the official implementation of the clustering-based approach proposed by (Zhang et al., 2025a). Query embeddings are derived from gte-Qwen2-7B-instruct (Li et al., 2023), with k-means clustering subsequently applied using $k = 64$. Notably, this method is non-parametric and involves no neural network training.

**Retrieval-based Router.** We extract the retrieval-based prior selection in SMACS (Tang et al., 2025) as a retrieval-based router, which is a variant of the KNN router. For a fair comparison, we utilize gte-Qwen2-7B-instruct (Li et al., 2023) to generate the embedding of each query. For hyperparameters, we set the base retrieval number as 50 and the tolerance threshold coefficient as 0.95.

**MoA.** To facilitate a fair comparison with other multi-LLM frameworks while minimizing computational overhead, we implement a single-layer MoA (Wang et al., 2024a) architecture. This setup consists of three LLMs as proposers to generate reference responses and one LLM as the aggregator for the final response. We employ standardized prompts across all

models to ensure consistent aggregation performance. Given that the vanilla MoA design lacks an inherent model selection mechanism, we conduct a stronger baseline by selecting three LLMs with high performance: DeepSeek-V3.2-Thinking, DeepSeek-V3.2-Speciale, and Qwen3-235B-A22B-Thinking-2507 to provide references, with DeepSeek-V3.2-Speciale acting as the final aggregator.

**Self Consistency.** To establish a competitive baseline while avoiding data contamination, we identify and select the best LLM for each dataset based on its training set performance. For each test query, this LLM is sampled four times to ensure a similar computational cost to other approaches. For benchmarks with well-defined ground truth, e.g., GPQA and AIME, we apply standard majority voting across the four responses. In contrast, for open-ended tasks such as Arena-Hard, we utilize gte-Qwen2-7B-instruct (Li et al., 2023) to generate an embedding for each candidate response, subsequently determining the final output through an embedding-based consensus mechanism.

**Majority Voting.** To comprehensively evaluate the efficacy of collective voting, we investigate two majority voting configurations: 1) a diverse ensemble of ten open-source models, and 2) a high-capability ensemble comprising DeepSeek-V3.2-Thinking, DeepSeek-V3.2-Speciale, and Qwen3-235B-A22B-Thinking-2507. The aggregation protocol remains consistent with the aforementioned self-consistency implementation, utilizing gte-Qwen2-7B-instruct (Li et al., 2023) as the underlying embedding model for consensus determination. Each LLM generates the response once with the same prompt and inference parameters, and votes for the final answer with the same weight.

**Symbolic-MoE.** For Symbolic-MoE (Chen et al., 2025a), we characterize the capabilities of each LLM using the same training dataset as JiSi. Specifically, Qwen3-235B-A22B-2507 is employed to annotate task-relevant keywords for each query. To facilitate keyword alignment between the test and training sets, we utilize gte-Qwen2-7B-instruct (Li et al., 2023) to compute similarity scores for the mapping process. While the proposer selection follows a pipeline identical to the original Symbolic-MoE, the aggregator selection is adapted due to the absence of a dedicated aggregation question bank. Instead, we dynamically select the LLM with the highest proficiency based on the competency profiling of the current test query. We follow the hyperparameter configurations, maintaining a softmax temperature of 0.5.

**Self-MoA.** For Self-MoA, we adopt a single-layer architecture to maintain architectural similarity with the MoA setup. Following the core paradigm of Self-MoA (Li et al., 2025b), we first identify the optimal LLM for a given test query based on its performance within the corresponding training datasets. This LLM then serves in a dual capacity as both the proposer and the aggregator. Specifically, the optimal LLM generates three independent inference passes, which it subsequently aggregates to produce the final response.

### D.4. Hyperparameters

In this section, we provide all the hyperparameters in JiSi. For experiments in Sec. 4, we use the same setting of hyperparameters without otherwise mentioning. Specifically, $K$, $N^{base}$, $\beta$, $\gamma$, $\epsilon$, $\sigma$, $\delta$, $t$ are set to 3, 50, 0.5, 0.95, 0.5, 0.3, 0.2, 0.8, respectively. To avoid extremely long context, the aggregatee number is truncated to 2 if the token sum of 3 aggregatees exceeds 13,000. For computing the cost, if the official API is available, we use the official API price. Otherwise, we adopt the stable API servers with the lowest price in OpenRouter.

### D.5. The Division Metric of Reference Accuracy Study

In the Sec. 4.5, we divide the test queries into three categories based on their difficulty. Specifically, "Easy" questions are defined as those that over 80% of the models can answer correctly. "Medium" questions require 30% $sim$ 80% of the models to respond correctly, while "Hard" questions require that less than 30% of models can generate correct responses.

## E. Similarity Analysis Cases

To intuitively demonstrate the superiority of our proposed query-response mixed similarity over the previous query-based similarity, we provide three qualitative examples in Fig. 8, Fig. 9, and Fig. 10. For each case, we present the similarity assessment results generated by different algorithms, accompanied by a detailed analysis of the query pair. These examples illustrate that while query-based similarity is limited to textual overlap and surface-level semantics, our proposed query-response mixed similarity more accurately captures difficulty information and the required domain knowledge.

## Similarity Analysis Case 1

Query-based Similarity: **75.47%**    ▶    **Query-Response Mixed Similarity (Ours): 22.49%**

### Query A: Elementary Level (LiveMathBench)

Solve the following math problem step by step. The last line of your response should only contain your final answer inside a \boxed{} command.

In a long line of people arranged left to right, the 1013th person from the left is also the 1010th person from the right. How many people are in the line?

Remember to put your final answer on the last line using the format \boxed{$ANSWER}.

### Query B: Olympiad Level (AIME)

Define $f(x) = ||x| - \frac{1}{2}|$ and $g(x) = ||x| - \frac{1}{4}|$. Find the number of intersections of the graphs of

$$y = 4g(f(\sin(2\pi x))) \quad \text{and} \quad x = 4g(f(\cos(3\pi y))).$$

Remember to put your final answer on the last line using the format \boxed{$ANSWER}.

**Summary:**
Despite nearly identical instructions and mathematical formatting, they reside in fundamentally different cognitive domains.
**Query A** is a basic logic and arithmetic problem based on the Inclusion-Exclusion principle. It requires only a single-step logical leap, identifying the person counted twice and basic addition/subtraction. The solution path is linear, deterministic, and demands minimal cognitive load.
**Query B** is an Advanced Functional Analysis problem typical of Olympiad-level mathematics. It involves multi-layered nested absolute value functions creating complex sawtooth geometries, coupled with the frequency analysis of composite trigonometric functions. Solving this necessitates sophisticated spatial visualization to count intersections of interwoven curves within a unit square—a task requiring high-order abstraction and precise symbolic manipulation.

*Figure 8.* The similarity analysis case 1.

## Similarity Analysis Case 2

Query-based Similarity: **87.80%**   ▶   **Query-Response Mixed Similarity (Ours): 29.76%**

### Query A: Inorganic Chemistry (GPQA)

Answer the following multiple choice question. The last line of your response should be of the following format: 'Answer: $LETTER' (without quotes) where LETTER is one of ABCD. Think step by step before answering.

Five binary compounds of fluorine with element Y are known. The bright-red substance A1 decomposes at 293 K into A2 ($\omega$F=31.96%) and fluorine. A1 oxidizes xenon under normal conditions. A3 can also interact with xenon. A3 is obtained by fluorination of element Y with fluorine. By adding Y in a 1:1 molar ratio to a hot colorless concentrated solution of A4, A5 can be obtained. In water, A5 decomposes with the formation of two substances. Indicate the range in which the molecular weight of the substance A4 falls.
A) 140-160    B) 220-240    C) 160-180    D) 110-130

### Query B: Theoretical Physics (GPQA)

Answer the following multiple choice question. The last line of your response should be of the following format: 'Answer: $LETTER' (without quotes) where LETTER is one of ABCD. Think step by step before answering.

In a parallel universe where a magnet can have an isolated North or South pole, Maxwell's equations look different. But, specifically, which of those equations are different?
A) The one related to the divergence of the magnetic field.
B) The ones related to the circulation of the electric field and the divergence of the magnetic field.
C) The one related to the circulation of the magnetic field and the flux of the electric field.
D) The ones related to the divergence and the curl of the magnetic field.

**Summary:**

The textual similarity is deceptively high due to identical prompt engineering and scientific nomenclature. However, the reasoning trajectories required for these tasks are fundamentally distinct.

**Query A** is an intensive *Inorganic Chemistry* question involving multi-step deductive reasoning. It requires identifying an unknown element through its various fluorides ($A_1$ to $A_5$), tracking redox properties with Noble gases, and calculating mass fractions. The cognitive load is dominated by domain-specific fact retrieval and stoichiometric calculations in a high-dimensional search space of chemical compounds.

**Query B**, conversely, belongs to the realm of Theoretical Electrodynamics. It challenges the solver to modify the classical Maxwellian framework under the hypothetical assumption of magnetic monopoles. The reasoning is first-principles-based, focusing on the symmetry of vector calculus operators, such as divergence and curl, and how they adapt to non-zero magnetic charges and currents. It requires conceptual synthesis of physical laws rather than multi-stage compound identification.

*Figure 9.* The similarity analysis case 2.

---

**Similarity Analysis Case 3**

Query-based Similarity: **69.56%**   ▶   Query-Response Mixed Similarity (Ours): **16.94%**

**Query A: Geometrical Optics (MMLU-PRO)**

Answer the following physics question. The last line of your response should be of the following format: 'Answer: $LETTER' (without quotes) where LETTER is one of ABCDEFGHIJ.

A solid glass sphere of radius R and index of refraction 1.50 is silvered over one hemisphere. A small object is located on the axis of the sphere at a distance 2R from the unsilvered surface. Find the position of the image formed by the refracting and reflecting surfaces.

A. 4R from the mirror surface     B. R from the spherical glass surface
C. 2R from the mirror surface     D. 2R from the spherical glass surface
E. 4R from the spherical glass surface     F. R/2 from the spherical glass surface
G. 5R from the spherical glass surface     H. R from the mirror surface
I. 3R from the spherical glass surface     J. 3R from the mirror surface
Let's think step by step.

**Query B: Ethical Philosophy (MMLU-PRO)**

Answer the following philosophy question. The last line of your response should be of the following format: 'Answer: $LETTER' (without quotes) where LETTER is one of ABCDEFGHIJ.

Feinberg claims that the best way to pursue happiness is to:
A. strive for success.     B. pursue knowledge.
C. pursue happiness.     D. focus on material wealth.
E. none of the above.     F. help others to be happy.
G. forget about happiness.     H. live in the moment.
I. avoid pain.     J. pursue pleasure.
Let's think step by step.

- - - - - - - - - - - - - - - - - - - - - - - - - - - - - - - - - - - - - -

**Summary:**
While both queries share the same multiple-choice template and "step-by-step" instructions, the underlying reasoning kernels are orthogonal.
**Query A** is a rigorous Geometrical Optics problem. Its solution requires executing a deterministic sequence of mathematical operations: applying the spherical surface refraction formula, followed by the mirror reflection formula, and a final refraction pass. It demands precise numerical computation and the ability to model light propagation within a complex physical system ($n = 1.50$, radius $R$).
**Query B**, in contrast, addresses the Paradox of Hedonism as proposed by Joel Feinberg. This is a Conceptual/Theoretical task that involves understanding the abstract philosophical argument that happiness is a byproduct of other pursuits rather than a direct goal. The reasoning path is based on interpretive knowledge of ethical theory rather than a calculated derivation.

*Figure 10.* The similarity analysis case 3.

## F. Aggregation and Routing Switch Cases

In this section, we present two intuitive examples in Fig. 11 and Fig. 12 to demonstrate how inferior information flows can mislead an aggregator that otherwise performs correctly in isolation, leading to incorrect outputs. To address this vulnerability, our proposed adaptive routing-aggregation switch, guided by prior scores, exhibits a higher sensitivity in detecting such inaccuracies. By selectively employing the more robust routing, our approach effectively and efficiently mitigates the impact of noise and enhances the overall system performance.

## Aggregation and Routing Switch Case 1

### Query: The "Xing" Mechanical Computer Challenge (HLE)

In a recent excavation of an ancient Chinese tomb, people found two books called Ching and Shu. The Ching contains exactly 9999 symbols of yinyang-wuxing, such as yin-water or yang-fire. The Shu contains around 3000 ancient Chinese characters. Using advanced language modeling and computer vision technologies, people discovered that the Shu is written with only 108 most common Chinese characters at that time. ... Then, they found that the Ching is actually the binary dump of a program written for an ancient Chinese mechanical computer called Xing and the Shu is its manual. Xing's architecture is similar to the von Neumann model. It uses the decimal system rather than binary, so each yinyang-wuxing symbol is a decimal digit (D). It has 10 registers indexed 0-9... It has a memory of 10000D, indexed 0000-9999. Locations 9999 and 9998 are mapped to input and output buffer. Code is loaded into memory at 0000... The program in the Ching has exactly 108 instructions. The remaining is data. This program will decode a secret message encrypted in the data into Chinese characters in the Shu. Answer K as the highest number of characters can be decoded from the Ching.

### Ground Truth

$$K = 4598$$

### Aggregator Output (DeepSeek-V3.2-Speciale) — *INCORRECT*

**Explanation:** The Ching consists of 9999 decimal digits. The program within it has exactly 108 instructions, each 6 digits long (opcode:1, register:1, operand:4), occupying $108 \times 6 = 648$ digits. The remaining $9999 - 648 = 9351$ digits are data. The program decodes this data into Chinese characters from a set of 108 symbols. To maximize the number of characters $K$, we treat the entire data as a single decimal integer $N$ (range 0 to $10^{9351} - 1$). By converting $N$ to base 108, each base-108 digit corresponds to one character. The number of base-108 digits of $N$ is $\lfloor \log_{108} N \rfloor + 1$. The maximum is achieved when $N = 10^{9351} - 1$ (all digits 9). Compute $\log_{108}(10^{9351}) = \frac{9351}{\log_{10} 108}$. Using $\log_{10} 108 \approx 2.03342375548$, we get $\frac{9351}{2.03342375548} \approx 4598.5$.

Since this is not an integer, $\lfloor \log_{108}(10^{9351} - 1) \rfloor = 4598$, giving $K = 4598 + 1 = 4599$. Indeed, $108^{4598} \approx 10^{9349.68} < 10^{9351} - 1$ and $108^{4599} \approx 10^{9351.72} > 10^{9351} - 1$, confirming that the largest possible number of characters is 4599. The fixed-length 3-digit encoding would yield only $9351/3 = 3117$ characters, which is not optimal.

**Answer:** 4599    **Confidence:** 95%

### Proposer References (Prior Scores & Performance)

1. **DeepSeek-V3.2-Speciale** (Prior Score: 0.357) — **CORRECT**
   *Explanation:* The Ching contains 9999 decimal digits. The program occupies $108 \times 6 = 648$ digits, leaving 9351 digits for data. The secret message is decoded into Chinese characters from an alphabet of size 108. To maximize the number of characters, we treat the data as a base-10 integer and convert it to base-108. The number of base-108 digits obtainable is the largest $L$ such that $108^L \leq 10^D$. With $D = 9351$, we have $L \leq \lfloor 9351/\log_{10}(108) \rfloor$. $\log_{10}(108) \approx 2.033423755$, so $9351/2.033423755 \approx 4598.5$, giving $L_{max} = 4598$.
   **Answer: 4598**                                                                 **Confidence:** 80%

2. **DeepSeek-V3.2-Thinking** (Prior Score: 0.282) — **INCORRECT**
   *Explanation:* The Ching contains 9999 decimal digits. The program consists of 108 instructions, totaling 648 digits. The remaining 9351 digits constitute the data segment. Since the Shu uses 108 distinct characters, each character requires at least 3 decimal digits for representation (as 2 digits only provide 100 possibilities). Assuming fixed-length encoding, the maximum number of characters that can be stored in the data is 9351 divided by 3, which equals 3117. This aligns with the mention that the Shu contains around 3000 characters.
   **Answer: 3117**                                                                 **Confidence:** 80%

3. **DeepSeek-R1-0528** (Prior Score: 0.080) — **INCORRECT**
   *Explanation:* To determine the highest number of characters $K$ that can be decoded from the Ching, we need to maximize the number of Chinese characters that can be extracted from the data segment. The data segment starts at address 648 and ends at address 9997 (since 9998 and 9999 are I/O buffers). The number of digits in the data segment is $9997 - 648 + 1 = 9350$. In base 108, the number of digits $K$ required to represent $N$ is given by $K = \lfloor \log_{108}(10^{9350}) \rfloor + 1$. Calculate $\log_{108}(10^{9350}) = \frac{9350}{\log_{10}(108)} \approx \frac{9350}{2.03342} \approx 4598.28$. Thus, $\lfloor 4598.28 \rfloor + 1 = 4599$. Therefore, $K = 4599$ is the maximum as $10^{9350} - 1$ can be encoded in 4599 base-108 digits.
   **Answer: 4599**                                                                 **Confidence:** 95%

**Summary:**
This case exemplifies the misleading inherent in multi-LLM aggregation. Despite the presence of a correct reference from proposer 1, the aggregator failed to aggregate the truth. It was swayed by the high-confidence yet flawed reasoning from proposer 3, which erroneously incremented the result via an incorrect ceiling/range logic ($4598 + 1$). Instead, a router based on the prior score will successfully solve this by assigning the task solely to the top-tier mathematical reasoning expert.

*Figure 11.* The aggregation and routing switch case 1.

## Aggregation and Routing Switch Case 2

### Query: Celebrity Fact-Verification

What day, month, and year was John Philip Bughaw born?

### Ground Truth

**November 7, 2008**

### Aggregator Output (Qwen3-235B-A22B-2507) — *INCORRECT*

**Analysis:** John Philip Bughaw, widely known by his stage name **"Balang"**, is a Filipino child dancer who gained international fame in 2015–2016 for his viral dance covers. While numerous online sources list his date of birth as **January 11, 2009**, this date cannot be verified through authoritative or primary sources.
**Critical Assessment:**
- **Contemporary News Reports**: credible media outlets in late 2015 described him as being 6 years old, supporting a birth year of 2009.
- **Lack of Official Confirmation**: There is no official public record or direct statement from his family.
- **Conflicting Claims**: Some sources list November 7, 2008, or December 28, 2004, which are treated as inconsistent or likely erroneous.
**Conclusion:** The most contextually supported information indicates he was born in **2009**, most likely around **early January 2009**. Therefore, while January 11, 2009, is commonly cited, it remains unverified.
**Answer:** | Born in 2009 (Exact date unverified) |     **Confidence:** High

### Proposer References (Prior Scores & Performance)

1. **Qwen3-235B-A22B-2507** (Prior Score: 0.840) — **CORRECT**
   John Philip Bughaw was born on **November 7, 2008**.

2. **GLM-4.6** (Prior Score: 0.433) — **INCORRECT**
   John Philip Bughaw was born on December 28, 2004. (Day: 28, Month: December, Year: 2004).

3. **Qwen3-235B-A22B-Thinking-2507** (Prior Score: 0.427) — **INCORRECT**
   *Reasoning:* I cannot provide the exact birth date with verified accuracy due to a lack of authoritative public records and conflicting online information. Media reports from 2015/2016 referred to him as 6 or 7 years old, strongly suggesting a birth year of 2009. While January 11, 2009, is frequently listed, it lacks primary verification. The only reasonably reliable information is that he was born in 2009.

**Summary:**
This case highlights a failure in knowledge synthesis under informational ambiguity. Although the expert with the highest prior score 0.84 explicitly provided the true birth date, the aggregator was swayed by the cautious reasoning from proposer 3 and the conflicting data from proposer 2.
The aggregator prioritized a plausible-sounding consensus (2009) derived from secondary media age-reporting over the specific, accurate fact provided by the most competent model. This confirms that for factual retrieval where the truth is sparsely documented but known to specific experts, a router that trusts the top-tier expert is more robust than an aggregator that attempts to reconcile conflicting narratives, which often results in the dilution of precise facts into generalized, unverified aggregation.

*Figure 12.* The aggregation and routing switch case 2.

