# OpenReview forum: "Beyond Gemini-3-Pro: Revisiting LLM Routing and Aggregation at Scale"
_ICML.cc/2026/Conference — ICML 2026 regular_

### Official Review · Reviewer_o6nq · 2026-02-22

**Soundness:** 2
**Presentation:** 3
**Significance:** 3
**Originality:** 2
**Overall Recommendation:** 3
**Confidence:** 3

**Summary:**

The paper studies collective intelligence with multiple LLMs, focusing on the classic tradeoff between: choose one model per query (efficient but may miss complementary strengths), versus combine outputs from multiple models (stronger but more costly/noisy).
The paper claim current methods hit bottlenecks because: (i) routers based only on query embeddings fail to capture difficulty and deeper semantics, (ii) aggregators are often fixed and not task-adaptive, and (iii) systems combine routing+aggregation in a static way rather than adapting per query.
They propose JiSi, a “minimalist route-and-aggregate” framework built on a pre-built embedding bank of training queries and model responses. JiSi has three key components:
1. Support-Set-based Aggregator Selection (SAS): choose an aggregator LLM using a coarse support set retrieved by query similarity, intended to balance domain-specific vs general aggregation ability.
2. Query-Response Mixed Routing (QRR): select candidate LLMs using not only query similarity, but also response embedding similarity and a proxy for difficulty via token-cost/length similarity; this is implemented by filtering the support set with a weighted combination of these criteria and computing fine-grained model priors.
3. Adaptive Routing–Aggregation Switch (ARS): based on fine-grained priors, dynamically decide whether to route to a single model or aggregate multiple (threshold-based), aiming to reduce unnecessary aggregation cost/noise.

Main headline results: JiSi achieves 72.15 avg across the nine tasks, surpassing their listed “Gemini-3-Pro” baseline at 71.00 avg, and is reported to save 53.23% cost versus Gemini-3-Pro in their cost accounting.

**Compliance With Llm Reviewing Policy:**

Affirmed.

**Final Justification:**

The rebuttal addressed some of my concerns but I believe the overall paper needs improvement and can be written better for later submissions.

**Key Questions For Authors:**

1. Add an amortized cost analysis including embedding bank construction cost/time, and show break-even points.
2. Strengthen scaling evidence with controlled experiments (random subsets, fixed-quality subsets).
3. Broaden evaluation beyond single-round SWE-bench (or clearly downscope the claim and rename the SWE-bench setting to avoid confusion).
4. Add more “modern” baselines if feasible: you already compare to several router and multi-LLM baselines, but it would help to include any recent selection methods not in RouterBench (if they exist in your timeframe), and/or a strong “best-of-K with verifier” baseline to isolate whether gains come from response-conditioned filtering vs just sampling multiple models. (You do include Self-Consistency and MoA, which is good, but there may be more recent selection/verifier-style approaches.)

**Strengths And Weaknesses:**

Strengths:
1. The paper identifies plausible, well-motivated weaknesses of query-only routing and fixed aggregation strategies, and proposes concrete mechanisms addressing each.
2. Strong empirical performance vs their baseline set, including both router baselines and multi-LLM aggregation baselines (MoA, self-consistency, majority voting, symbolic MoE, self-MoA).
3. Cost reporting, analysis section is informative.

Weakness:
1. Encapsulating three innovations in one paper is a overstretch. It will be good to showcase each point in separate papers with detailed explanation of their use.
2. They repeatedly describe JiSi as training-free, but the system depends on building an embedding bank that requires: collecting model responses for all training questions for all M models, and computing a per-model capability vector from correctness indicators against labels.
Even if no gradient-based training happens, this is still a substantial offline supervision + compute requirement (and not obviously available in real deployments, where ground-truth labels for user queries are often unavailable).
The cost table appears to focus on inference/evaluation cost, but does not account for offline bank construction cost
3. The fine-grained filtering uses response embeddings and token costs from candidate models. Concretely, they retrieve a coarse support set, choose top-K candidate models, then query all K models to get responses and costs, then compute response similarity / cost similarity to filter the support set and compute refined priors.
That means: even if the final switch chooses Nfm=1 (routing), the system has already paid for K model calls to compute the response-dependent priors (unless an early-exit variant exists, which is not clearly described).
4. Evaluation protocol limitations (especially SWE-bench + judge models). The evaluation is explicitly non-interactive single-round. For SWE-bench, they “pack all context in a single query” because aggregation methods struggle with multi-turn/tool usage.
This is a large deviation from the standard SWE-bench paradigm (iterative patching, tool execution, etc.). The reported SWE-bench numbers may not translate to real SWE-bench settings.
5. The paper suggests a shift from neural scaling laws to “multi-LLM collaboration scaling law.” But the demonstrated scaling is from 5 to 10 models, added in chronological order (which also likely correlates with “models get better over time”). A better scaling study would include: random subsets of fixed size, controlled inclusion of weaker/stronger models, or hold average quality constant while varying count.

---

> ### Author Rebuttal · Authors · 2026-03-31
>
> **Q1:** Packing three innovations is an overstretch; they should be separate papers.
>
> **A1:** Our three innovations are not isolated but highly coupled, fundamentally driven by information refinement and noise reduction. Query-Response Mixed Routing selects high-quality sources; the Adaptive Switch filters noise; and Support-Set-based Selection identifies the optimal aggregator. Exploring them independently remains promising future work.
>
> **Q2:** "Training-free" ignores offline embedding bank construction costs. Provide an amortized cost analysis and break-even points.
>
> **A2:** Building the bank via open-source APIs costs ~\\$ 670 offline, vastly cheaper than model training. JiSi saves ~$0.02 per query compared to Gemini-3-Pro, meaning the break-even point is reached at just 33.5K queries, a trivial volume for deployed LLM systems. The economical efficiency comes from:
>
> 1. **One-Time Cost:** The initial embedding bank is built offline and serves all future requests with almost no adding online latency or costs.
>
> 2. **Scalability:** Expanding the bank requires merely appending new entries, completely avoiding retraining.
> Meanwhile, for label independence, without ground-truth labels, JiSi can seamlessly utilize verifiers, LLM-as-a-judge, or user feedback for embedding bank construction.
>
> **Q3:** Fine-grained filtering computes $K$ responses even if routing to a single model ($N_{fm}=1$), causing extra overhead.
>
> **A3:** We acknowledge this overhead, but it is a necessary investment for routing precision. As shown in **Table 1** and **Section 4.4**, response-aware routing significantly outperforms query-only routing. Moreover, over 70% of test queries actually trigger aggregation ($N_{fm}>1$), fully utilizing these $K$ calls. In future work, we plan to use cheaper, smaller models (e.g., 8B) as probes to generate preliminary responses, further minimizing overhead.
>
> **Q4:** Evaluation on SWE-bench is a non-interactive single-round, deviating from the standard iterative paradigm. Rename it to avoid confusion.
>
> **A4:** We agree. While **Table 1** currently includes a footnote, we will explicitly rename this metric to **SWE-Single-Round** to eliminate any confusion. This setup follows the LLMRouterBench protocol and is designed to rigorously test models' tool-free code reasoning and extreme long-context capabilities.
>
> **Q5:** The scaling study confounds model count with temporal quality improvements. Provide controlled experiments holding average quality constant.
>
> **A5:** We conducted a new controlled scaling experiment, progressively adding models while strictly limiting the average model pool performance gap to within 1.82. As shown in **Table t11**, the scaling curve and upward trend remain consistent even when controlling for average model quality, validating the multi-LLM collaboration scaling law.
>
> **Table t11**: Controlled scaling study.
> ||Pool Avg|AIME|Arena-Hard|GPQA|HLE|LiveCodeBench|LiveMathBench|MMLU-Pro|SimpQA|SWE-Bench|Reasoning&Knowledge|Coding&Engineering|Chat&Factuality|
> |---|---|---|---|---|---|---|---|---|---|---|---|---|---|
> |5|55.71|83.3|83.33|83.33|18.67|76.03|75.68|85.33|49.15|32.67|69.26|54.35|66.24|
> |6|56.92|88.98|84.44|90|24.38|77.6|78.38|86.22|49.15|30.67|73.59|53.35|66.80|
> |7|55.1|88.89|84.44|90|24.69|80.44|78.38|86.89|49.38|31.33|73.77|54.47|66.91|
> |8|56.41|94.44|84.22|86.67|27.93|86.12|75.67|87.11|50.85|39.33|74.36|59.89|67.54|
> |9|55.33|94.44|85.33|86.67|27.47|87.07|75.68|87.11|50.15|39.33|74.27|63.20|67.74|
> |10|55.39|94.44|88.44|86.67|27.62|89.27|81.08|86.78|53.7|41.33|75.32|65.30|71.07|
>
>
> **Q6:** Add modern baselines like "best-of-K with verifier" or LLM-Blender to isolate where gains come from.
>
> **A6:** We have expanded our baselines to include more verifier-based methods [1,2,3,4]. We applied repeated sampling on the best single model per dataset to create the strongest possible baselines. As shown in **Table t12**, JiSi consistently outperforms these approaches, confirming our gains stem from the elegant coupling and multi-step filtering of routing and aggregation, rather than mere repeated sampling.
>
> **Table t12:** Comparison with other baselines.
> |Method|AIME|Arena-Hard|GPQA|HLE|LiveCodeBench|LiveMathBench|MMLU-Pro|SimpQA|SWE-bench|Avg|
> |---|---|---|---|---|---|---|---|---|---|---|
> |LLM-Blender|94.44|83.11|88.33|28.55|90.54|75.68|87.33|54.7|36|70.96|
> |MAV|94.44|82.89|86.67|30.56|90.85|75.68|86.78|54.78|35.33|70.89|
> |BoK(LLM-judge)|94.44|82.89|88.33|27.31|90.22|75.68|87|55.16|34.67|70.63|
> |BoK(RM)|94.44|86.89|85|25.93|88.64|75.68|87.67|52.7|40.67|70.85|
> |**JiSi(Ours)**|94.44|88.44|86.67|27.62|89.27|81.08|86.78|53.7|41.33|**72.15**|
>
> [1] Llm-blender: Ensembling large language models with pairwise ranking and generative fusion.
>
> [2] Multi-Agent Verification: Scaling Test-Time Compute with Multiple Verifiers.
>
> [3] Learning to summarize from human feedback.
>
> [4] Judging LLM-as-a-Judge with MT-Bench and Chatbot Arena.

---

> > ### Author Rebuttal · Reviewer_o6nq · 2026-04-03
> >
> > Thanks for the clarification. I appreciate the effort to perform additional experiments to answer my questions. I will stick my scores as I believe that the paper can be better written.

---

> > > ### Author Response · Authors · 2026-04-03
> > >
> > > Thanks for your response and for taking the time to review our rebuttal. We are very grateful that you appreciate our efforts and the additional experiments we provided.
> > >
> > > We are glad that our new controlled scaling experiments (**Table t11**) and the expanded baseline comparisons against verifier-based methods (**Table t12**) successfully addressed your technical questions regarding the scaling study and baselines. We are also pleased that the amortized cost analysis clarified the economic efficiency and break-even point of JiSi.
> > >
> > > Regarding the presentation and writing, we completely respect your perspective. We are fully committed to polishing the manuscript further to ensure it meets the highest standards of ICML. We will ensure these are rigorously addressed in the camera-ready version.
> > >
> > > Thank you again for your constructive feedback.

---

### Official Review · Reviewer_pfRd · 2026-03-06

**Soundness:** 3
**Presentation:** 4
**Significance:** 2
**Originality:** 3
**Overall Recommendation:** 4
**Confidence:** 3

**Summary:**

The paper proposes JiSi, a training-free “route-and-aggregate” framework for orchestrating multiple open-source LLMs to surpass strong monolithic systems. JiSi introduces three main ideas: (i) Query-Response Mixed Routing that augments query embeddings with embeddings of candidate responses and a response-length proxy for difficulty; (ii) Support-Set-based Aggregator Selection that balances domain specificity and general aggregation skill using a larger retrieved support set; and (iii) an Adaptive Routing-Aggregation Switch that prunes weak responses and falls back to single-model routing when appropriate. Experiments on nine benchmarks with a pool of ten open-source LLMs report state-of-the-art average performance and substantial cost reductions versus strong closed-source baselines.

**Compliance With Llm Reviewing Policy:**

Affirmed.

**Key Questions For Authors:**

1. How are K and the threshold t chosen, and how sensitive are results to these choices across datasets? Please provide sensitivity curves and per-dataset optimal values.
2. For open-ended tasks evaluated with LLM-as-judge, what steps were taken to mitigate judge bias (e.g., multi-judge aggregation, calibration prompts, randomized formatting)? Can you report agreement metrics or robustness to judge swaps?
3. Have you conduct repeated experiments? I noticed the variances are not reported, which affects the soundness of the results, especially in Table-1.

**Limitations:**

yes

**Strengths And Weaknesses:**

The paper tackles an important and timely systems question: can multi-LLM orchestration outperform strong monolithic models at lower cost? JiSi’s three components are thoughtfully designed and, in conjunction, appear to deliver consistent gains in the reported experiments. The training-free, retrieval-centered design is appealing for modularity and ease of adoption.

One limitation is the practicality of the routing mechanism. The response-aware routing requires decoding responses from a top-K candidate set before selecting the final model output, which introduces additional computational overhead and may blur the distinction between routing and ensembling. This design can increase latency and potentially create long-tail response times in real deployments. In addition, the heuristic used to estimate task difficulty based on response length may not generalize well across tasks, since verbosity and stylistic variation can affect length independently of difficulty.

The experimental evaluation also leaves several important questions unanswered. The variances are not reported, which makes it unclear if the improvements come from stable outperformance or variance between runs. The paper also does not report end-to-end latency or wall-clock runtime, which is particularly important for a routing system that generates multiple candidate responses. Furthermore, sensitivity analyses are limited: key hyperparameters such as the candidate set size K, selection thresholds, and weighting coefficients are not systematically studied, and the reported improvements are relatively small without accompanying variance or statistical significance analysis.

Additionally, several aspects of the presentation and positioning make the work harder to evaluate. The mathematical formulation seems unnecessarily complicated, and uses non-standard notation and occasionally opaque operators, which makes the method difficult to verify and reproduce.

---

> ### Author Rebuttal · Authors · 2026-03-31
>
> **Q1:** Response-aware routing introduces decoding overhead, blurring routing and ensembling. End-to-end latency is not reported.
>
> **A1:** We acknowledge this overhead, but it is deliberately designed to bridge routing and ensembling, and this preliminary decoding is crucial for accurate routing. If aggregation is triggered, these preliminary responses are directly reused to prevent redundant computation. As detailed in response to Reviewer Fjes (**Tables t1 & t2**), the results confirm the overhead is controlled and entirely acceptable given the significant performance gains.
>
> **Q2:** Estimating difficulty based on response length may not generalize due to verbosity and stylistic variations.
>
> **A2:** Our length-aware heuristic is robust and generalizes well for three key reasons:
> 1. **Intra-Model Comparison:** Cost similarity is calculated exclusively **within** the same model's responses, eliminating cross-model verbosity/stylistic bias.
> 2. **Empirical Generalizability:** As demonstrated in Section 4.4 and Table 3, incorporating length(cost) similarity reduces routing deviation across multiple diverse datasets, proving its broad applicability.
> 3. **Intrinsic Property:** For LLMs, greater difficulty inherently requires more compute (tokens) for a given LLM [1][2]. JiSi leverages this intrinsic property.
>
> **Q3:** Variances are not reported. Are improvements from stable outperformance or run-to-run variance?
>
> **A3:** Results from 5 independent runs (**Table t7**) show marginal variance. Mean performance remains significantly higher than baselines, demonstrating JiSi's improvements are highly stable and statistically sound.
>
> **Table t7:** Independent repeated experiments.
> |AIME|Arena-Hard|GPQA|HLE|LiveCodeBench|LiveMathBench|MMLU-Pro|SimpleQA|SWE-bench|ALL|
> |---|---|---|---|---|---|---|---|---|---|
> |94.44±0.0|88.49±0.64|87.00±1.39|27.58±0.26|89.34±0.35|81.08±0.0|86.8±0.12|53.7±0.14|41.2±0.99|72.18±0.14|
>
> **Q4:** Limited sensitivity analysis for $K$ and threshold $t$. How are they chosen? Please provide per-dataset optimal values.
>
> **A4:** Extensive sensitivity tests (**Tables t4-t6** in response to Reviewer Fjes) confirm JiSi is robust to hyperparameter variations. **Tables t8-t9** detail per-dataset results and optimal settings for $K$ and t, showing consistently superior performance across wide ranges. For the setting in JiSi, $K$ was determined via coarse grid search in route-only mode; t was set empirically.
>
> **Table t8:** Ablation study on candidate set size $K$.
>
> |K|AIME|Arena-Hard|GPQA|HLE|LiveCodeBench|LiveMathBench|MMLU-Pro|SimpleQA|SWE-bench|Avg|
> |---|---|---|---|---|---|---|---|---|---|---|
> |3|94.44|88.44|86.67|27.62|89.27|81.08|86.78|53.7|41.33|72.15|
> |5|94.44|88.44|86.67|27.78|89.59|78.38|87.44|53.39|42.67|72.09|
> |7|94.44|86.67|86.67|27.93|87.7|78.38|87.44|53.39|41.33|71.55|
> |10|94.44|85.56|86.67|27.93|88.01|78.38|87.44|53.39|38.67|71.17|
>
> **Table t9:** Ablation study on switch threshold $t$.
> |t|AIME|Arena-Hard|GPQA|HLE|LiveCodeBench|LiveMathBench|MMLU-Pro|SimpleQA|SWE-bench|Avg|
> |---|---|---|---|---|---|---|---|---|---|---|
> |0.5|94.44|87.78|86.87|29.94|89.59|81.08|87.11|51|41.33|72.13|
> |0.6|94.44|87.28|86.67|29.94|90.22|81.08|87.11|51.39|40.67|72.09|
> |0.7|94.44|86.83|86.67|29.32|89.27|81.08|87.71|50.85|41.33|71.94|
> |0.8|94.44|88.44|86.67|27.62|89.27|81.08|86.78|53.7|41.33|72.15|
> |0.9|94.44|86.83|85|27.62|85.17|75.68|87.11|54.85|43.33|71.11|
>
> **Q5:** Mathematical formulation is overly complicated and uses non-standard notation, hindering reproducibility.
>
> **A5:** We will simplify mathematical notations to make core innovations accessible. For full reproducibility, we will open-source the codebase and pre-built embedding bank upon acceptance.
>
> **Q6:** How did you mitigate judge bias for open-ended tasks? Are there agreement metrics or robustness tests for judge swaps?
>
> **A6:** For our LLM-as-a-judge evaluation, we strictly followed the standard single-judge protocol in [3], which is also applied in [4] [5]. To further validate robustness against judge bias, we conducted multi-judge experiments (using DeepSeek V3, o3-mini, Gemini 2.5 Flash compared with DeepSeek V3 in single-judge) on Arena-Hard and SimpleQA. As shown in **Table t10**, swapping to a multi-judge setup yields minimal performance variance, and the relative ranking of methods remains unchanged, confirming the high robustness of our evaluation.
>
> **Table t10:** Different LLM-as-judge setups.
> |Method|Arena-Hard|SimpleQA|
> |---|---|---|
> |Gemini-3-Pro(Single)|74.55|70.03|
> |Gemini-3-Pro(Three)|73.33|70.03|
> |Symbolic-MoE(Single)|73.11|52.7|
> |Symbolic-MoE(Three)|72.67|52.7|
> |JiSi(Single)|88.44|53.7|
> |JiSi(Three)|87.78|53.7|
>
> [1] Learning How Hard to Think: Input-Adaptive Allocation of LM Computation.
>
> [2] Scaling LLM Test-Time Compute Optimally Can be More Effective than Scaling Parameters for Reasoning.
>
> [3] LLMRouterBench: A Massive Benchmark and Unified Framework for LLM Routing.
>
> [4] The Llama 3 Herd of Models.
>
> [5] Qwen2 Technical Report.

---

> > ### Author Rebuttal · Reviewer_pfRd · 2026-04-02
> >
> > Thank you. I will maintain my already positive score.

---

> > > ### Author Response · Authors · 2026-04-03
> > >
> > > Thank you very much for reviewing our rebuttal and maintaining your positive evaluation of our work.
> > >
> > > We deeply appreciate your time, support, and the constructive feedback you provided during the review process. Your insights have been highly valuable in helping us polish the manuscript.
> > >
> > > Thank you again for your time and expertise!

---

### Official Review · Reviewer_Fjes · 2026-03-12

**Soundness:** 3
**Presentation:** 3
**Significance:** 3
**Originality:** 3
**Overall Recommendation:** 4
**Confidence:** 3

**Summary:**

The paper aims to explore collective intelligence through multi-LLM collaboration as a viable alternative to scaling monolithic models. And the paper proposes JiSi, a training-free, scalable framework that orchestrates multiple open-source LLMs to surpass the performance of proprietary giants like Gemini-3-Pro. Extensive experiments across 9 challenging benchmarks demonstrate that coordinating 10 open-source LLMs via JiSi outperforms Gemini-3-Pro. However, I still have some questions, please see the weakness.

**Compliance With Llm Reviewing Policy:**

Affirmed.

**Key Questions For Authors:**

See weakness

**Strengths And Weaknesses:**

Strength:

* The framework successfully beats state-of-the-art closed-source models (including Gemini-3-Pro and Claude 4.5) across diverse domains. The overall idea is good.
* The paper evaluates the system on 9 very difficult and modern benchmarks, supported by solid ablation studies, cost analysis, and qualitative case studies that clearly explain why the method works.

Weakness:

* Inference Latency Overhead: While the monetary cost is well-analyzed, the paper lacks a detailed discussion on latency. Because JiSi requires candidate models to generate "preliminary responses" to compute the fine-grained prior score before the final aggregation, the sequential wall-clock time might be noticeably higher than querying a single monolithic model.
* Dependency on the Embedding Bank: As a non-parametric method, JiSi heavily relies on the quality, density, and diversity of the pre-built Embedding Bank. The paper does not thoroughly explore how the system performs when faced with extreme Out-Of-Distribution queries that have no highly similar counterparts in the bank.
* Hyperparameter Sensitivity: The framework introduces a fair number of hyperparameters (e.g. $\epsilon, \delta$ for similarity weighting, threshold t, base number N_{base}). Although they are fixed in the experiments, tuning these manually for different deployment scenarios or smaller/larger candidate pools might be non-trivial in practice.

If the author can address the issues, I will be happy to increase the score.

---

> ### Author Rebuttal · Authors · 2026-03-31
>
> **Q1:** Lack a detailed discussion on latency. Generating preliminary responses might cause noticeable wall-clock time compared to monolithic models.
>
> **A1:** We will add a latency analysis. As shown in **Table t1**, JiSi's ~1.7x latency overhead vs. Gemini-3-Pro is an acceptable trade-off given its substantial cost reduction (save ratio = 53.23%) and performance gains (+1.15). Compared to multi-LLM baselines, JiSi offers competitive latency and superior performance via three designs:
>
> 1. **Adaptive Routing-Aggregation Switch:** By dynamically routing many queries directly to a single model, JiSi bypasses full aggregation, drastically reducing average inference time compared to static multi-LLM methods.
>
> 2. **Parallel Inference for Preliminary Responses:** Candidate models are queried simultaneously during this phase. The time cost is bottlenecked only by the slowest single LLM pass, rather than sequential accumulation.
>
> 3. **Response Caching:** If the selected aggregator matches a model from the preliminary phase, JiSi directly reuses the cached response. As shown in **Table t2**, this tightly controls the incremental time cost from preliminary generation to final aggregation.
>
> **Table t1:** Average latency, cost, and performance.
> |Method|Avg Latency(s)|All Cost(\\$)|Avg Perf(%)|
> |---|---|---|---|
> |Gemini 3 pro|151.46|135.46|71.00|
> |MoA|225.74|62.09|67.58|
> |Self Consistency|221.03|78.57|70.66|
> |Self-MoA|303.67|80.47|66.52|
> |Symbolic-MoE|275.67|65.69|66.89|
> |**JiSi(Ours)**|**261.63**|**63.36**|**72.15**|
>
> **Table t2:** Latency breakdown.
> |Stage|Avg Latency(s)|
> |---|---|
> |Preliminary|206.01|
> |Preliminary+Supply|211.78|
> |Preliminary+Supply+Aggregation|261.63|
>
>
> **Q2:** JiSi relies on the pre-built Embedding Bank. It may fail when facing extreme OOD queries.
>
> **A2:** We acknowledge this valid concern. However, JiSi remains robust for two key reasons:
>
> 1. **Robustness via Semantic Retrieval:** Even without exact matches, partial semantic overlap within the bank allows the support set to roughly estimate query difficulty and domain requirements, preventing catastrophic routing failures. We validated this by isolating AIME, GPQA, and LiveCodeBench as an extreme OOD test set, building the bank exclusively from the remaining datasets. **Table t3** shows that despite a slight drop compared to the InD setting, JiSi maintains high performance and clearly outperforms baselines.
>
> 2. **Continual Scalability:** As a training-free framework, JiSi can seamlessly and cost-effectively incorporate new query types into the embedding bank, continually enhancing its OOD resilience over time without computationally expensive model retraining.
>
> **Table t3:** Performance comparison on OOD settings.
>
> |Method|AIME|GPQA|LiveCodeBench|Avg|
> |---|---|---|---|---|
> |Majority Voting|88.89|86.67|85.80|87.12|
> |MOA|94.44|78.33|88.33|87.03|
> |Symbolic-MoE|94.44|80.00|85.80|86.75|
> |Self Consistency|77.78|55.00|60.58|64.45|
> |**JiSi(OOD)**|**94.44**|**83.33**|**87.38**|**88.38**|
> |**JiSi(InD)**|**94.44**|**86.67**|**89.27**|**90.13**|
>
> **Q3:** The framework introduces multiple hyperparameters. Manually tuning these might be non-trivial.
>
> **A3:** Empirical evidence demonstrates that JiSi is highly robust to hyperparameter variations within reasonable ranges, allowing a single set of parameters to generalize well across diverse scenarios:
>
> 1. **Cross-Task Generalization:** Across nine distinct task categories, JiSi utilized an identical hyperparameter set and achieved superior performance, indicating deployment-specific tuning is generally unnecessary.
>
> 2. **Scale Insensitivity:** In our scaling experiments (Section 4.3), expanding the model pool from 5 to 10 models with the exact same hyperparameters yielded consistent gains, confirming robustness to candidate pool variations.
>
> 3. **Stable Ablation Results:** Extensive ablation studies (**Table t4, t5, t6**) show minimal performance fluctuations across varying query similarity weights ($\epsilon$), switch thresholds ($t$), and base support set sizes ($N^{base}$), confirming the framework's overall stability.
>
>
>
> **Table t4:** Ablation study on query similarity weight $\epsilon$.
>
> |$\epsilon$|0.3|0.4|0.5|0.6|0.7|
> |---|---|---|---|---|---|
> |Avg Perf|71.04|72.18|72.15|71.47|70.95|
>
>
> **Table t5:** Ablation study on adaptive switch threshold $t$.
>
> |$t$|0.5|0.6|0.7|0.8|0.9|
> |---|---|---|---|---|---|
> |Avg Perf|72.13|72.09|71.94|72.15|71.11|
>
> **Table t6:** Ablation study on base support set size $N^{base}$.
>
> |$N^{base}$|20|30|50|70|100|
> |---|---|---|---|---|---|
> |Avg Perf|70.69|71.87|72.15|71.07|71.28|

---

> > ### Author Rebuttal · Reviewer_Fjes · 2026-04-01
> >
> > I thank the authors for their rebuttal. They addressed some of my concerns. However, regarding W2, Table T3 shows that in some OOD settings, JiSi does not perform particularly well compared with the baselines, which suggests that the method may still depend on the quality of the Embedding Bank.
> >
> > Regarding W3, I would also encourage the authors to provide more insight into why different hyperparameter settings lead to these behaviors. For example, it appears that a smaller $t$ performs better, but the underlying reason is not clearly explained. It would also be helpful to discuss what happens if $t$ is reduced even further.
> >
> > For these reasons, I will keep my score unchanged.

---

> > > ### Author Response · Authors · 2026-04-02
> > >
> > > We sincerely thank you for your prompt rebuttal acknowledgement and your continued recognition of our work. We appreciate the opportunity to further clarify the mechanisms behind JiSi.
> > >
> > > **Regarding W2 (OOD Performance and Embedding Bank Quality):** We acknowledge that the quality of the embedding bank is critical to JiSi, as it directly impacts the system's ability to accurately evaluate model adaptability for a given query. The Table t3 was to verify that our method is not overly sensitive to OOD queries, avoiding catastrophic failure, and that it can still maintain an overall advantage compared to baseline methods. However, we also want to highlight a key practical advantage of our framework: unlike approaches that require expensive model retraining, our embedding bank can be continuously and sustainably expanded at a low cost. In real-world deployments, this flexibility means that the embedding bank can be dynamically updated with new queries, progressively and efficiently mitigating OOD issues over time.
> > >
> > > **Regarding W3 (Insights into Hyperparameter $t$):** Thank you for encouraging a deeper discussion on the underlying behaviors of our hyperparameters. To explain the ablation results for the adaptive switch threshold $t$: As the threshold $t$ increases, noise and inferior model responses are progressively filtered out. This purifies the information flow and helps to consistently improve the system's aggregation performance (peaking around $t = 0.8$). However, when $t$ becomes excessively high, the strict filtering inadvertently eliminates various high-quality reference responses. This causes a drastic drop in the diversity of the expert responses provided to the aggregator. Diverse and high-quality responses are the fundamental prerequisites for effective aggregation. When diversity is overly suppressed, the aggregation process loses its efficacy, and the entire system gradually degenerates into a router (relying on a single response). This degradation is the underlying reason for the overall performance drop at very high thresholds.
> > >
> > > Conversely, if $t$ is reduced even further (e.g., below 0.5), the threshold becomes too lenient. While this maximizes diversity, it also allows excessive noise and low-quality responses to pass through to the aggregator. As demonstrated in our "Aggregation and Routing Switch Cases" (Appendix F), incorporating inferior information easily misleads the aggregator, which would inevitably degrade overall performance. Therefore, a balanced threshold is necessary to ensure noise is filtered while critical diversity is preserved.
> > >
> > > We hope these explanations address your remaining concerns. Thank you once again for your constructive engagement with our work.

---

### Official Review · Reviewer_E7GQ · 2026-03-12

**Soundness:** 3
**Presentation:** 2
**Significance:** 3
**Originality:** 3
**Overall Recommendation:** 4
**Confidence:** 4

**Summary:**

This paper proposes an LLM ensemble method with new designs for both the "routing stage" and the "post-inference aggregation stage".

**Compliance With Llm Reviewing Policy:**

Affirmed.

**Key Questions For Authors:**

Please refer to the above.

**Limitations:**

yes

**Strengths And Weaknesses:**

**Strengths**

1) I believe the proposed method will be effective.
The designs of several components in the method are reasonable.

**Weaknesses and Suggestions**

1) The introduction is not clear enough. The overall readability of the introduction needs to be strengthened.

2) Related to the previous point, for Section 3, I suggest that the authors place the "analysis of the method's advantages" after the methodology introduction. That is, start the descriptions of sections 3.1 and 3.2 as early as possible. I do not recommend keeping the lead-in text before Section 3.1 in its current position.

3) Additionally, I suggest presenting Figure 4 before Figure 3.

4) For the Related Work section, suggest adding the highly relevant survey paper ("Harnessing Multiple Large Language Models: A Survey on LLM Ensemble") and including the "ensemble-during-inference" methods mentioned in that survey.
Additionally, suggest adding the the introduction of related works including methods similar to the proposed approach as well as classic methods, e.g.,  LLM-Blender.

---

> ### Author Rebuttal · Authors · 2026-03-31
>
> **Q1:** The introduction is not clear enough. The overall readability of the introduction needs to be strengthened.
>
> **R1:** We sincerely thank the reviewer for the constructive feedback. In the revision, we will restructure the introduction to significantly enhance its readability and logical flow. Specifically, we will streamline the discussion on the limitations of current monolithic scaling to transition more directly into the motivation for our collective intelligence framework. Furthermore, we will explicitly map the three core bottlenecks we identified in existing methods, including query-based paradigm limitations, static aggregation, and underutilized complementarity, directly to our three corresponding innovations.
>
> **Q2:** Related to the previous point, for Section 3, I suggest that the authors place the "analysis of the method's advantages" after the methodology introduction. That is, start the descriptions of sections 3.1 and 3.2 as early as possible. I do not recommend keeping the lead-in text before Section 3.1 in its current position.
>
> **R2:**  We agree that introducing the method earlier will improve the reader's comprehension of the framework. In the revision, we will reorganize Section 3 to immediately present the Framework Overview (currently Section 3.1) and the Embedding Bank (currently Section 3.2). The current introductory text, which details our rethinking of existing routers, aggregation, and combination strategies, will be relocated to follow the methodology description. Placed there, it will serve as an analytical summary that highlights the specific advantages of our approach compared to the baselines.
>
> **Q3:** Additionally, I suggest presenting Figure 4 before Figure 3.
>
> **R3:** We carefully considered the placement of these figures during drafting. Figure 3 is intentionally positioned first to provide an intuitive, high-level conceptual comparison between existing routing/aggregation paradigms and our approach. Our intention is to establish a clear conceptual motivation, showing exactly why our three innovations are necessary, before diving into the granular, operational details of the pipeline presented in Figure 4. However, we recognize that this transition might feel abrupt to some readers. To address this in the revision, we will add clearer transitional sentences linking the high-level concepts in Figure 3 directly to their implementations in the pipeline shown in Figure 4, leading to a smoother and more logical reading experience without losing the initial conceptual motivation
>
> **Q4:** For the Related Work section, suggest adding the highly relevant survey paper ("Harnessing Multiple Large Language Models: A Survey on LLM Ensemble") and including the "ensemble-during-inference" methods mentioned in that survey. Additionally, suggest adding the introduction of related works, including methods similar to the proposed approach as well as classic methods, e.g., LLM-Blender.
>
> **R4:** We thank the reviewer for highlighting this important literature. In the revised related Works section, we will incorporate the suggested survey paper[1] and enrich our discussion with a dedicated subsection on "ensemble-during-inference" methods such as GAC[2], DEEPEN[3], etc.. Meanwhile, we will also include more classic methods, e.g., LLM-Blender[4].
>
> [1] Harnessing Multiple Large Language Models: A Survey on LLM Ensemble.
>
> [2] Breaking the ceiling of the llm community by treating token generation as a classification for ensembling.
>
> [3] Ensemble learning for heterogeneous large language models with deep parallel collaboration.
>
> [4] Llm-blender: Ensembling large language models with pairwise ranking and generative fusion.

---

> > ### Author Rebuttal · Reviewer_E7GQ · 2026-04-02
> >
> > Thanks. I have carefully read the response. The authors actively adopted my suggestions, and I maintain my original judgment. I suggest that the authors open-source the code, as it would be meaningful to the community.

---

> > > ### Author Response · Authors · 2026-04-03
> > >
> > > Thank you very much for your careful reading and maintaining a positive evaluation of our work. We deeply appreciate your constructive suggestions, which have been instrumental in improving the quality of our manuscript.
> > >
> > > We completely agree with your perspective on the importance of contributing to the community. We will release our codebase and the embedding bank upon the acceptance of this paper. We hope our open-source release will facilitate future research in multi-LLM collaboration.
> > >
> > > Thank you again for your valuable feedback and support!

---

### Decision · Program_Chairs · 2026-04-30

**Decision:**

Accept (regular)

**Comment:**

After reviewing the four submissions, I recommend a weak accept for this paper. The work addresses an important and timely problem—orchestrating multiple open-source LLMs to rival proprietary models—and proposes a well-motivated, training-free framework with three thoughtful components. Reviewers consistently acknowledge the technical soundness, strong empirical results across nine challenging benchmarks, and solid ablation and cost analyses. However, several weaknesses temper the enthusiasm. These include insufficient discussion of inference latency overhead and practical deployment considerations, reliance on an embedding bank whose construction cost is not fully accounted for, limited sensitivity analysis for key hyperparameters, and the lack of variance reporting and statistical significance tests. Additionally, the evaluation on SWE-bench deviates from standard protocols, and the paper’s three innovations feel somewhat overpackaged. Despite these limitations, the core contributions are solid, the reported performance gains are convincing, and the work is likely to be built upon by others. With moderate revisions addressing the experimental rigor and practical trade-offs, the paper would be significantly enhanced.